# The emergence of magnetic ordering at complex oxide interfaces tuned by defects

D.-S. Park [1✉], A. D. Rata[2], I. V. Maznichenko[2], S. Ostanin[2], Y. L. Gan [3], S. Agrestini[4,5], G. J. Rees [6], M. Walker [7], J. Li[8], J. Herrero-Martin [4], G. Singh[9], Z. Luo [7], A. Bhatnagar[2,10], Y. Z. Chen [3], V. Tileli [8], P. Muralt[8], A. Kalaboukhov[9], I. Mertig[2], K. Dörr[2], A. Ernst[11,12] & N. Pryds [3]

Complex oxides show extreme sensitivity to structural distortions and defects, and the intricate balance of competing interactions which emerge at atomically defined interfaces may give rise to unexpected physics. In the interfaces of non-magnetic complex oxides, one of the most intriguing properties is the emergence of magnetism which is sensitive to chemical defects. Particularly, it is unclear which defects are responsible for the emergent magnetic interfaces. Here, we show direct and clear experimental evidence, supported by theoretical explanation, that the B-site cation stoichiometry is crucial for the creation and control of magnetism at the interface between non-magnetic $ABO_3$-perovskite oxides, $LaAlO_3$ and $SrTiO_3$. We find that consecutive defect formation, driven by atomic charge compensation, establishes the formation of robust perpendicular magnetic moments at the interface. Our observations propose a route to tune these emerging magnetoelectric structures, which are strongly coupled at the polar-nonpolar complex oxide interfaces.

[1] Group for Ferroelectrics and Functional Oxides, Swiss Federal Institute of Technology—EPFL, 1015 Lausanne, Switzerland. [2] Institut für Physik, Martin-Luther-Universität Halle-Wittenberg, 06120 Halle, Germany. [3] Department of Energy Conversion and Storage, Technical University of Denmark, DK-2800 Kgs Lyngby, Denmark. [4] ALBA Synchrotron Light Source, Cerdanyola del Vallès, 08290 Barcelona, Spain. [5] Oxford Diamond Light Source, DidcotOxford OX11 0DE, UK. [6] Department of Materials, University of Oxford, Oxford OX1 3PH, UK. [7] Department of Physics, University of Warwick, Coventry CV4 7AL, UK. [8] Institute of Materials, Swiss Federal Institute of Technology—EPFL, 1015 Lausanne, Switzerland. [9] Department of Microtechnology and Nanoscience—MC2, Chalmers University of Technology, SE-412 96 Gothenburg, Sweden. [10] Zentrum für Innovationskompetenz SiLi-nano®, 06120 Halle, Germany. [11] Max Planck Institute of Microstructure Physics, 06120 Halle, Germany. [12] Institute of Theoretical Physics, Johannes Kepler University, 4040 Linz, Austria. ✉email: dspark1980@gmail.com

Versatile, but intriguing electronic and magnetic phenomena, such as two-dimensional high-mobility electron gas[1,2], magnetoelectricity[3,4], chiral magnetic domains[5,6] and topological phenomena[7,8], have all been observed in various strongly correlated complex oxide systems with tunable magnetoelectric properties. These physical systems have significant potential in spin-based energy applications, such as low-energy consumption electronics. A representative complex oxide system is the interface of non-magnetic LaAlO$_3$ and SrTiO$_3$ (LAO/STO), which shows non-trivial ferromagnetism persisting up to room temperature (RT) and associated magnetoelectric coupling effects[9–13]. In order to disentangle the observed magnetoelectric interactions, several predications have been made to describe these phenomena, such as an induced crystal field splitting near the interfaces and orbital reconstruction of the composing transition metal (titanium, Ti 3d-bands) octahedra[12–16], anion defect-mediated exchange[17,18] and Rashba spin–orbit coupling[19,20]. Interestingly, after so many years of research in these interfaces, the fundamental origin of the localized magnetic moment is still under debate. The quasi-two-dimensional electron gas (q-2DEG) of LAO/STO with a high carrier density, above $\sim 1 \times 10^{13}$ cm$^{-2}$, typically shows very weak magnetism. Such weak magnetism in these buried interfaces is difficult to measure by conventional magnetometers, but possible with a scanning superconducting quantum interference device (SQUID) magnetometer[21]. Also, the magnetism has not been found to be dependent on the character of the interface, that is, n- or p-type[14]. Minute amount of magnetization with moment values ($m$) in the range of 0.001–0.3$\mu_B$/ unit cell (u.c.), $m \leq 2 \times 10^{-5}$ emu, have so far been reported using indirect magnetoelectric approaches[10,19–22]. The observations of the magnetoelectric interactions at the interfaces provide an opportunity to use these interfaces in spintronic applications that require electric field-tunable properties.

The formation of quantum-confined q-2DEG in LAO/STO heterostructures is characterized by chemically abrupt and homogeneous interfaces. The homogeneity is essential to obtain a polar/nonpolar interface arising from the stacking of the A-site cation (La$^{3+}$), populated in the first layer of LAO next to highly susceptible TiO$_2$-terminated STO[23]. This results in a large polar discontinuity driven by the induced electrostatic potential across the interface and indicated by the finite conductivity at the interface, which is observed only for atomic ratios, La/Al $\leq 0.97 \pm 0.3$[24]. In contrast, a p-type interface with a 2D hole gas was evoked in an STO/LAO/STO stacking heterostructure, eliminating the positively charged anion point defects[25]. In general, the highly conducting n-type interfaces practically never show strong magnetism ($m < 1 \times 10^{-5}$ emu)[9,10,26,27] and exhibit rather weak magnetoelectric signatures. A magnetic signature at the Ti site at the interface has been directly identified by surface-sensitive X-ray magnetic circular dichroism (XMCD)[15]. The most prominent magnetism was reported by Ariando et al.[10] for LAO–STO interfaces exhibiting a large disorder at the cation sites. The observed ferromagnetism at the interface, grown at a high oxygen partial pressure ($P_g$), persists up to RT. The formation of all possible energetically favourable cation defects and the related atomic intermixing and their influence on the interfacial magnetism in the LAO/STO system was proposed theoretically[28]. These studies indicate that the stoichiometry of the LAO over-layer and the subsequent ionic and/or electronic modification at the interface could be the key reasons for creating and modulating the interfacial magnetism. Figure 1a presents a collection of magnetic moments taken from the literature for the LAO–STO interfaces at different pulsed laser deposition (PLD) conditions. The observed trend shows that relatively weak interfacial magnetism ($m \leq \sim 2$ µemu) commonly occurs with a low oxygen pressure ($P_g \leq 10^{-4}$ mbar) growth atmosphere and a high laser

fluence ($E_g > 1.2$ J/cm$^2$) film growth, independent of the film thickness and size of the grown samples.

In this work, we provide deep insights into the yet unexplored effect of B-site cation defects on magnetism at the LAO/STO interface. Here, we establish a direct and controllable relation between the B-site cation vacancies (both Al and Ti vacancies) of the LAO/STO system and the induced magnetism. Our experimental and theoretical results clearly show that the two B-site cation vacancies and anti-site defects are the main sources for the creation and control of the interfacial magnetism. Formation of the B-site cation defects is correlated simultaneously, giving rise to a consecutive atomic charge compensation at the LAO/STO interface. It is found that the preferentially out-of-plane aligned defect dipoles have a significant impact on the magnetism and the correlated magnetotransport properties in this complex oxide interface system.

## Results

**Cation stoichiometry and magnetism at LAO/STO interfaces.** The effect of the LAO stoichiometry on the interfacial magnetism was studied by deposition of 9-u.c.-thick LAO layers on TiO$_2$-terminated STO (001) substrates using PLD. Details of the PLD thin film growth process are given in the "Method" section and Supplementary Fig. 1. After film growth at 700 °C, the samples were cooled down to RT in O$_2$ ($P_O = 200$ mbar). The thickness and quality of the grown films, as well as their stoichiometry, were studied and controlled by varying the laser fluence (energy per unit area) (see Fig. 1b). The laser fluency and plume–background gas interaction both significantly affect the transfer of ablated species to the surface of underlying substrates during deposition, resulting from the noncongruent ablation and/or mass-dependent variations in the atom and ion trajectories through the laser plume[29,30]. The results indicate that preferential scattering of light plume species result in La-rich LAO films under relatively low laser fluencies ($\sim 0.6$ J/cm$^2$). Conversely, a less effective scattering results in Al-rich films driven by higher laser fluencies (up to $\sim 1.8$ J/cm$^2$). Cation stoichiometry of ABO$_3$ perovskite with 50% A- and 50% B-site cations is taken as the baseline of the cation stoichiometry of the grown films. The high quality of layer-by-layer growth of the LAO films after deposition on STO was confirmed by atomic force microscopy and reflection high-energy electron diffraction (RHEED) (Supplementary Fig. 1). Figure 1b shows the atomic composition of the as-grown LAO/STO samples, determined by a surface-sensitive (a take-off angle (TOA) of 30°) X-ray photoelectron spectroscopy with effective probing depths of a maximum of $\sim 3.2$–5.6 nm for all the constituent element core levels of LAO and STO (see the "Methods" section and Supplementary Note 1). The results show that the ratio of A- and B-site cations in the 9-u.c. LAO layers is laser fluence dependent. No signatures from magnetic impurities were observed in these samples (see Supplementary Fig. 2).

Figure 1c shows the magnetic hysteresis loops measured at 5 K for the 9-u.c. LAO/STO samples as a function of laser fluence. The measured hysteresis loops were corrected by subtracting the constant diamagnetic and paramagnetic background, caused by an instrument's accessory (Supplementary Fig. 3). The saturation magnetic moment of the grown films gradually increases with the reduction in the laser energy fluence, that is, the reduction of the concentration of Al correlated with the increase in the La concentration (Fig. 1b). This indicates that either excessive La or reduced Al content plays a crucial role in creating and modulating the magnetism. Owing to the oxidation conditions of the samples, we expect negligible or no contribution of oxygen vacancies to the sample's magnetization. Our results directly reveal that the observed tunable magnetism of the LAO/STO

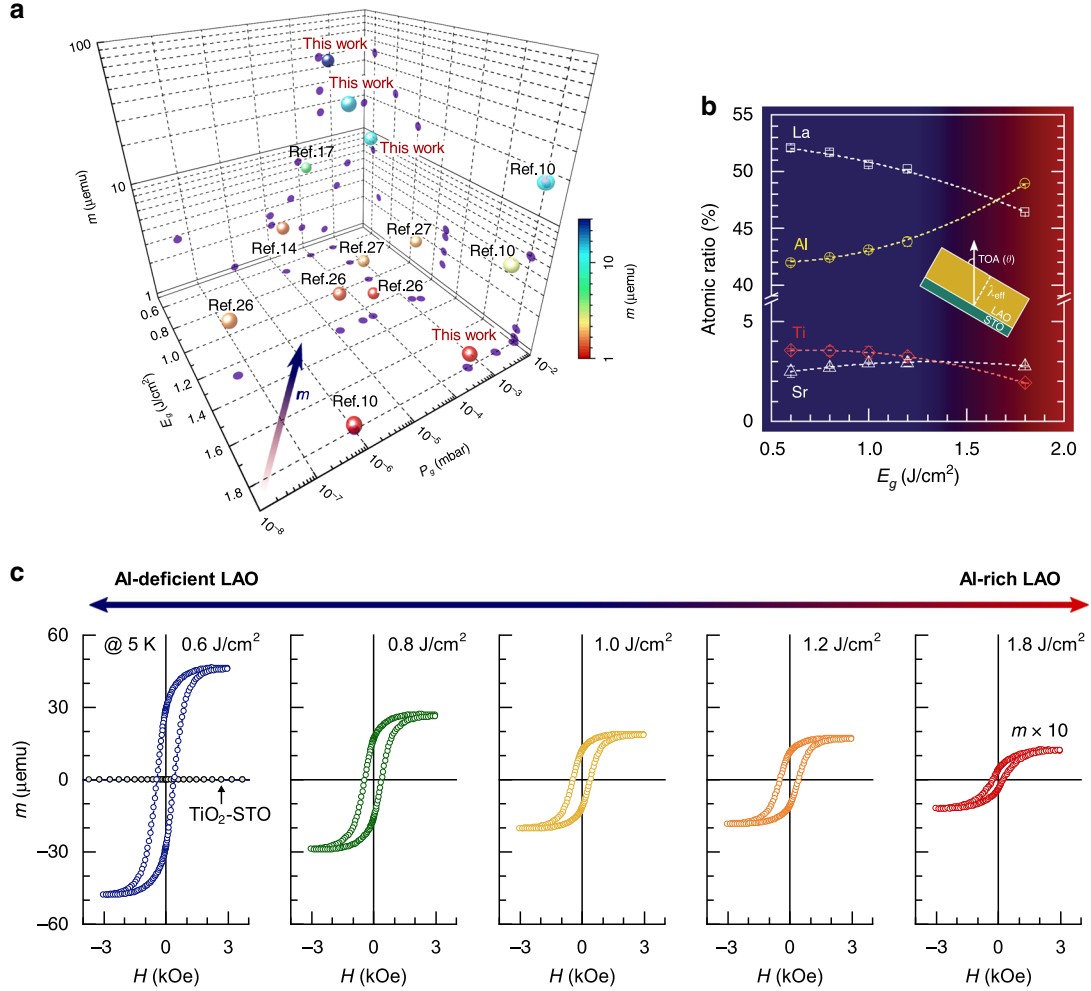

**Fig. 1 Stoichiometry dependence of magnetism in the LAO/STO interface. a** A comparison from the literature of the obtained saturation magnetic moment ($m$) of LAO/STO interfaces grown by different film growth parameters, that is, laser fluence ($E_g$) and oxygen partial pressure ($P_g$) using PLD: 10 u. c. LAO/STO (~5 × 5 mm²) from ref. [10], 5 u.c. LAO/STO (~5 × 5 mm²) from ref. [14], 10 u.c. LAO/STO (~5 × 5 mm²) from ref. [17], 24 u.c. LAO/STO from ref. [26], 5 u.c. LAO/STO from ref. [27] and 9 u.c. LAO/STO (~5 × 5 mm²) from this work. **b** The atomic ratio of La, Al, Ti and Sr of the grown 9-u.c. LAO/STO samples as a function of laser fluence $E_g$, determined by a surface-sensitive XPS. The insert shows a schematic of the X-ray photoemission geometry given by $\lambda_{eff} = \lambda \sin\theta_{TOA}$, where $\lambda_{eff}$ and $\theta_{TOA}$ are the effective inelastic mean free path of the photo-emitted electrons and emission angle with respect to the surface normal, respectively. **c** 5 K magnetic hysteresis loops of the 9-u.c. LAO/STO samples as a function of $E_g$.

system is primarily dependent on the cation stoichiometry of the LAO layer, and so is predominantly attributed to the effect of B-site cation (Al) deficiency.

To further clarify whether the magnetic moment of the nonstoichiometric LAO/STO samples is evoked solely by the Al vacancy content, an Al-deficient LAO layer (created using a laser fluence of ~0.6 J/cm²) was grown on a single crystalline LAO (001) substrate using identical growth conditions. As observed in Fig. 2a, at 5 K the saturation magnetic moment of the homoepitaxial sample (9 u.c. LAO/LAO(001)) is ~21% of the moment of the Al-deficient 9-u.c. LAO/STO heterostructure. This implies that a substantial part of the total moment measured in the Al-deficient LAO/STO system arises from the underlying STO layers. We have thus also prepared homoepitaxial films of 9 u.c. STO, deposited on TiO₂-STO(001) substrates under varying laser fluencies to qualitatively examine any effect of nonstoichiometry on the upper STO layers (Fig. 2d), although different kinetics of arriving species and resultant defect density occur during film growth[29]. Surprisingly, the saturation magnetic moment of nonstoichiometric STO layers at 5 K also strongly depends on laser fluence (Fig. 2d). It can be inferred from these results that

the enhanced magnetism in the Al-deficient LAO/STO system has apparent contributions from the B-site cation defects of both LAO and STO.

In order to understand the origin of magnetism and the contribution of B-site cation deficiency, we performed first-principles calculations utilizing density functional theory (DFT). The calculation details are given in the "Methods" section. To isolate the individual effect causing the B-site vacancy defects in these host systems, we separately calculated tetragonally distorted LAO (cubic STO) model structures embedding an Al (Ti) vacancy. Our calculations indeed show that the B-site cation vacancies create holes on acceptor-like energy states near the O 2$p$-dominated valence band of both the LAO and STO (Fig. 2b, c, e, f). Interestingly, both cation vacancies lead to spin polarization due to partially filled/unsaturated $p$-orbitals of the six nearest O ions. Such magnetic complexes were first predicted theoretically and later observed experimentally, for example, in ZnO[31–33]. The induced magnetization of ~70% is distributed into the six neighbouring O atoms. The total magnetic moment of the spin-polarized 2$p$ electrons of the O atoms surrounding the isolated Al (Ti) vacancy, in a relaxed tetragonal LAO (cubic STO), yields

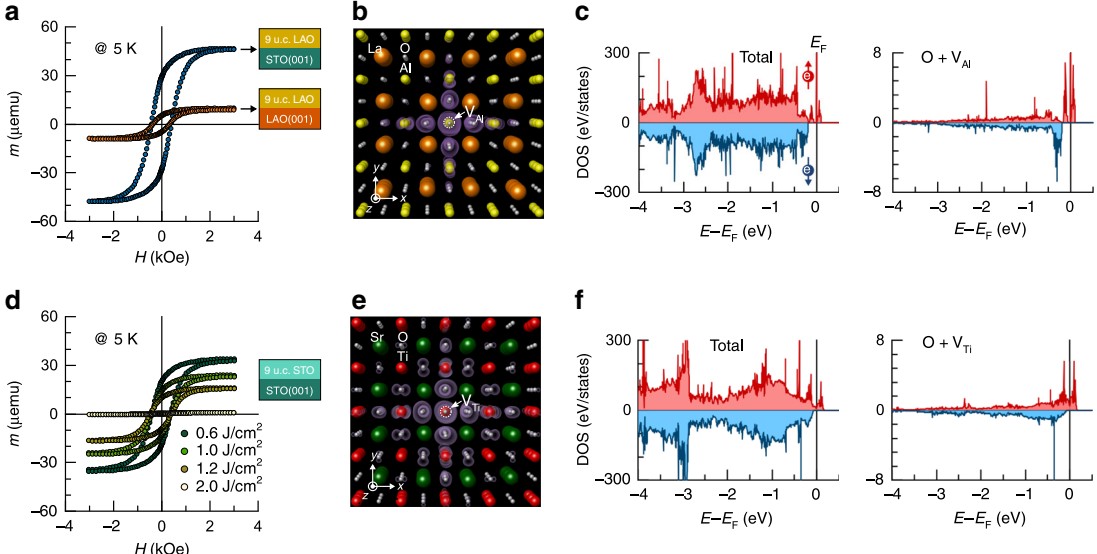

**Fig. 2 Origin of magnetism in B-site cation-deficient LAO/STO system. a** Comparison of the 5 K magnetic hysteresis loops of the B-site cation-deficient 9-u.c. LAO layers, grown on $TiO_2$-terminated STO(001) and LAO(001) substrates using a low laser fluence, $E_g$ = ~0.6 J/cm². **b** Magnetization density of a tetragonally distorted LAO with a $V_{Al}$. **c** The total spin-polarized density of states near the valence band states for the tetragonally distorted LAO with a $V_{Al}$ (in the left panel) and the corresponding local density of states of the neighbouring six oxygen atoms next to a $V_{Al}$ (in the right panel). **d** Laser fluence dependence of the 5 K magnetic hysteresis loops of 9-u.c. STO layers, grown on $TiO_2$-terminated STO(001) substrates. **e** Magnetization density of a cubic STO with a $V_{Ti}$. **f** Total density of-states near the valence band states for the STO with a $V_{Ti}$ (in the left panel) and the corresponding local density of states of the six neighbouring oxygen atoms next to a $V_{Ti}$ (in the right panel).

~$0.87\mu_B$/$V_{Al}$ ($0.48\mu_B$/$V_{Ti}$). Other possible vacancy defects such as $V_{Sr}$ and $V_O$ were also considered to contribute to the experimentally observed magnetism in this heterosystem. However, none of them showed the creation of a noticeable magnetic moment of STO in the oxygen-rich environment (see Supplementary Note 2 and Supplementary Fig. 4). Therefore, the DFT results affirm that the $V_{Al}$ and $V_{Ti}$ B-site vacancies are the only source for oxygen spin polarization in LAO and STO, respectively.

**B-site cation defect-induced magnetic ordering across the interface.** In addition to the tunable magnetization originating from $V_{Al}$ and $V_{Ti}$ at the B site in LAO and STO samples, thickness dependence of the magnetization in Al-deficient LAO/STO samples was also investigated at a fixed laser fluence of 0.7 J/cm². The saturation magnetization in the samples continuously increases from 2 u.c. LAO up to 9 u.c. LAO and eventually levels off at ~12 u.c. (Fig. 3a and Supplementary Fig. 5). The cation vacancy-mediated magnetism is indicative of an interfacial atomic relaxation across the interface, while the thickness-dependent magnetism is driven by cation thermodynamics at the interface and/or the saturation of vacancy defect level in the system as dictated by the amphoteric defect model[34]. Our DFT calculations show that the cation vacancies energetically tend to segregate at the interface (see Supplementary Fig. 6). Moreover, since no magnetic response was obtained from an annealed $TiO_2$-terminated STO substrate (Fig. 1c, left panel), we anticipate that only a single effect exists, caused by the $V_{Al}$ distribution throughout the LAO/STO system. Therefore, in these heterostructures, the creation of $V_{Ti}$ defects on the STO side of the heterointerface is induced by the presence of $V_{Al}$ in the LAO overlayer, as a consequence of the atomic charge compensation process. Outward diffusion of Ti atoms and/or the subsequent anti-site defect formation across the interface consequently creates $V_{Ti}$-mediated magnetism on the STO side. This was further confirmed by atomic-resolution scanning tunnelling electron microscopy (STEM), electron energy loss spectroscopy (EELS)

and energy-dispersive X-ray spectroscopy, as shown in Fig. 3b–d, illustrating the defect ordering across an Al-deficient LAO/STO interface. The anti-site defect can be rigidly coupled with the neighbouring cation vacancies along [001] due to lack of lateral Ti-O-Ti orderings. Our DFT results further indicate that the $Ti_{Al}$ anti-site defect in the tetragonally strained LAO layer creates a total magnetic moment of ~$1\mu_B$/$Ti_{Al}$: the $Ti_{Al}$ has itself a magnetic moment of ~$0.49\mu_B$ and the moment of the neighbouring La (the six nearest oxygen) atoms is $0.015\mu_B$ ($0.40\mu_B$) (Fig. 3e, f).

To verify the magnetic nature of the B-site cation-deficient LAO/STO system, O K-edge XMCD spectra for an oxidized 9-u.c. LAO/STO sample, grown by a laser fluence of 0.6 J/cm², is separately measured with applied fields of B = +6 and −6 T. A very clear field swap of the XMCD signal at the O K-edge, presented in Fig. 4a–c, demonstrates its magnetic nature. Three different contributions to the XMCD spectrum can be distinguished. The two contributions at 529 and 532–533 eV are seen where O 2p orbitals hybridized with Ti 3d $t_{2g}$ and $e_g$ orbitals, respectively, are known to occur[15]. Note that this magnetic signal cannot be due to $Ti^{3+}$ ions, produced by $V_O$ defects, because the spectral shape of the Ti $L_{2,3}$ XMCD spectrum, measured on the oxidized film (Supplementary Note 3 and Supplementary Fig. 7c), is very similar to the pure $Ti^{4+}$ spectrum, reported by Salluzzo et al.[17]. Moreover, the O K-edge XMCD can not only be attributed to $d_{xy}$-type ferromagnetism[15], since our result also reveals polarization of the oxygen states mixed with the Ti $e_g$ bands and other hybridized bands of the cations at higher energies. Thus, we conclude that the two XMCD contributions at 529 and 532–533 eV reflect the polarization of the oxygen bands, induced by both of $Ti_{Al}$ and $V_{Ti}$ defects. The XMCD signal centred at 535.5 eV occurs in a region, dominated by O 2p-states hybridized with La 5d, Al 3s and Al 3p empty bands and is associated with $V_{Al}$ defects. The observed O spin polarization nature remains at RT as a temperature-independent characteristic (Supplementary Fig. 7d). As a result, it is confirmed that the consecutive atomic charge compensation at the interface establishes magnetic orders from aligned cation vacancy–anti-

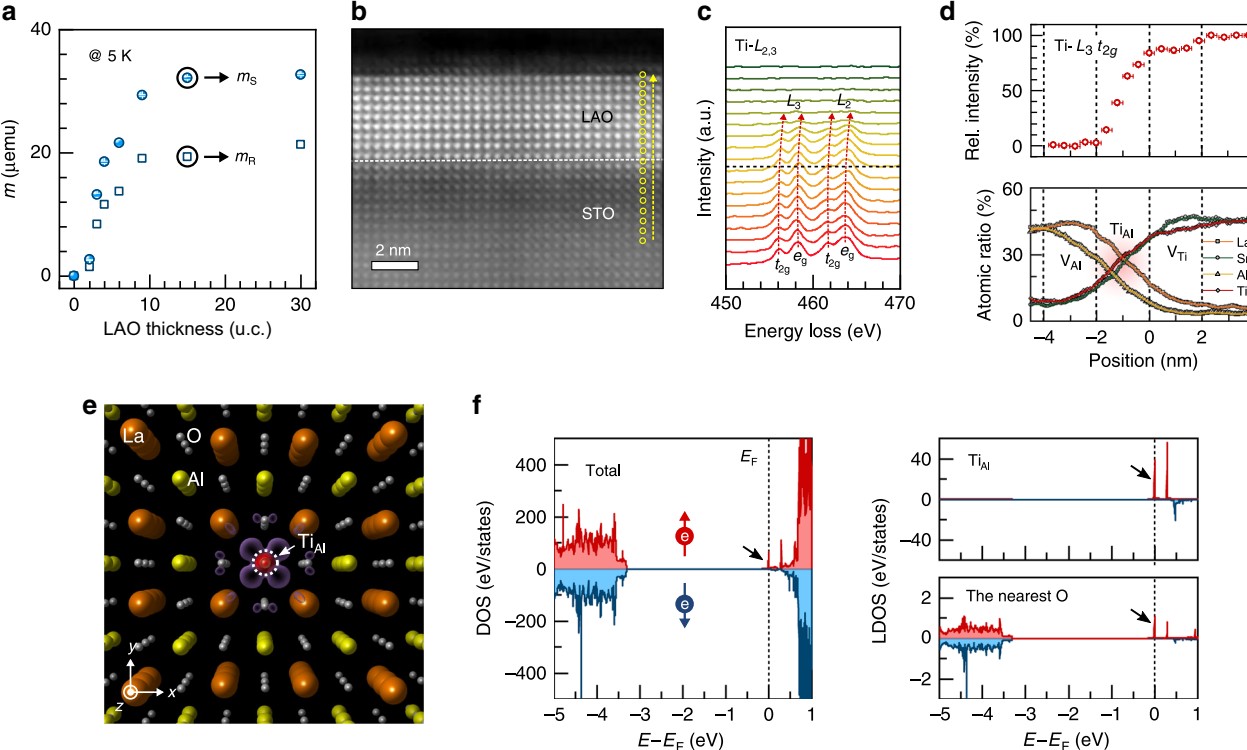

**Fig. 3 B-site cation defect distribution at the interface. a** Values of saturation magnetic moment ($m_S$) and remnant moment ($m_R$), as a function of the LAO thickness, varying from 2 to 30 u.c. at $E_g = 0.7$ J/cm². **b** Atomic-resolution high-angle annular dark-field STEM image of a B-site cation-deficient LAO/STO sample, projected along the <100> direction. The interface is denoted by a dashed line. **c** Ti $L_{2,3}$-edge EELS spectra for the LAO/STO heterostructure, collected from each layer position and denoted by the yellow circles in **b**. **d** Relative integrated intensity of the Ti $L_3$ $t_{2g}$ EELS spectra (in the upper panel). The values represent the relative Ti distribution across the interface (0 nm), obtained by dividing the $t_{2g}$ peak area of the Ti $L_3$-edge of each layer to that of the bottom-most STO layer, normalized to be 100%. The EDS line scan profile for the atomic ratios of La, Al, Sr and Ti, recorded from the top-most surface of the LAO/STO heterostructure (in the lower panel). **e** Magnetization density of a tetragonally distorted LAO with a $Ti_{Al}$. **f** The total spin-polarized density of states of the tetragonally strained LAO with a $Ti_{Al}$ (in the left panel) and the corresponding local density of states of the $Ti_{Al}$ and six neighbouring oxygen atoms next to a $Ti_{Al}$ (in the right panel).

site–vacancy defects as schematically illustrated in Fig. 4d. This might further reflect the enhanced delocalization of the magnetization density outward the B-site cation defect clusters as shown by DFT calculations. In other words, the total magnetic response of the B-site cation-deficient heterostructure results from an extended defect configuration at the interface, instead of isolated defects or individual layer contributions. Furthermore, the preferential orientation of the magnetic moment of the B-site cation-deficient LAO/STO system is revealed to be the out of plane in [001] direction (Fig. 4e) and temperature-independent between 5 and 300 K (Fig. 4f). The observed large ferromagnetism could be related to exchange interaction of the ordered defects with a large spin wave stiffness[35,36] and/or a collective planar orbital magnetism[37].

**Interplay between the B-site cation defect-induced magnets and q-2DEG.** We provide further evidences of magnetic ordering due to coexisting cation and anion defects from the observation of anomalous Hall effect (AHE) in magnetotransport measurements. Figure 5b–d shows the variable temperature sheet resistance, carrier concentration, and electron mobility of three different samples: a conventional B-site cation-rich LAO/STO sample ($E_g = 1.8$ J/cm²) without oxidation (cooled down to RT in the same deposition pressure, $P_g$) after film growth (see the "Methods" section and Supplementary Fig. 8a), and two B-site cation-deficient LAO/STO samples (0.6 J/cm²), both with and without oxidation. For the oxidized B-site cation-deficient LAO/STO sample, which possesses a large magnetic moment (Fig. 1c),

an electrical insulating behaviour is observed after eliminating the oxygen vacancies from the interface. This is due to either the lack of charge carriers and/or the absence of conducting channels because of localized carriers around the cation defects (see Supplementary Fig. 7b). In contrast, the B-site cation-rich ($E_g = 1.8$ J/cm²) and B-site cation-deficient (0.6 J/cm²) LAO/STO samples without oxidation show n-type behaviour characteristic of LAO/STO system. Below 50 K, there is a crossover from the linear ordinary Hall effect (LOHE) at a low magnetic field to the non-linear ordinary Hall effect (NOHE) at a high magnetic field. The NOHE originates by the presence of two types of charge carriers from different conduction bands with $d_{xy}$ and $d_{yz}/d_{xz}$ character. For conductive LAO/STO interfaces, the multiband conduction usually originates from the low-temperature suppression of the interband scattering for both the light and heavy mass band electrons in the STO crossing the Fermi level[38–41]. We fitted the low-temperature NOHE in both samples using a two-band model assuming different carrier densities ($n_1$ and $n_2$) and mobilities ($\mu_1$ and $\mu_2$) together with the AHE (see Supplementary Note 4 and Supplementary Fig. 9). In the linear region (one band), the B-site cation-deficient sample shows ~2.2 times lower sheet carrier density of $2.3 \times 10^{13}$ cm⁻², compared with that of the conventional LAO/STO one ($5.3 \times 10^{13}$ cm⁻²). This is due to the expected partial charge compensation effect between the coexisting p-type carriers (induced by cation vacancies) and n-type carries at high temperatures, $T > 100$ K. Interestingly, only in the nonlinear region, the derived low-mobility ($\mu_1$) and carrier density ($n_1$) of both samples show the expected discernable transport

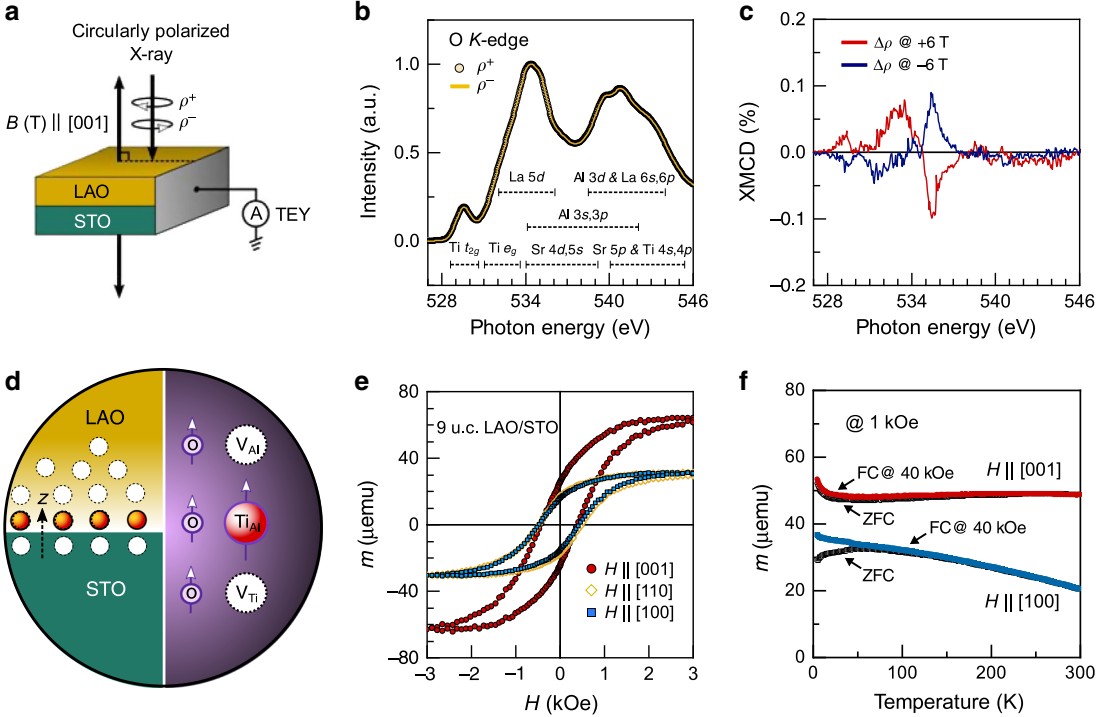

**Fig. 4 B-site cation defect-induced magnetic ordering at the interface. a** Schematics of X-ray magnetic circular dichroism (XMCD) measurement setup. Total electron yield (TEY) detection is obtained through excitation by using circularly polarized light ($\rho^+/\rho^-$) in normal incidence while applying an external magnetic field along the out-of-plane <001> directions. **b** A 20 K X-ray absorption spectra at the O $K$-edge of a B-site cation-deficient 9-u.c. LAO/STO sample, grown by $E_g = 0.6$ J/cm². The energy positions of the O-hybridized bands are estimated by the theoretical electronic structure calculations of the bulk LAO and STO. **c** O $K$-edge XMCD spectra, measured by swapping the applied fields, $B = +6$ and $-6$ T. **d** Schematics of consecutive atomic charge compensation via Ti out-diffusion and the resulting defect ordering and formation of magnetic moments across the interface. **e** A 5 K Magnetic hysteresis loops along [100], [110] and [001] crystallographic directions of the B-site cation-deficient 9-u.c. LAO/STO sample. **f** Temperature vs. magnetic moment for the 9-u.c. LAO/STO sample, measured at a magnetic field of 1 kOe along the out-of-plane [001] and in-plane [100] directions. FC represents the sample that is cooled down from RT to 5 K applying a field of 40 kOe, while ZFC represents a sample that is cooled down to 5 K without any field applied.

characteristics, while the corresponding high mobility ($\mu_2$) and carrier densities ($n_2$) are almost identical. This indicates that the high-mobility 2DEG is not significantly affected by defect scattering even in the B-site cation-deficient LAO/STO interface with a large density of locally distributed cation defects.

Another remarkable feature of the Hall effect is the presence of an additional low-field nonlinearity and hysteretic behaviour that originates from the AHE[39,42,43]. The AHE is emerging at low temperatures below 2 and 5 K for both the B-site cation-rich and B-site cation-deficient samples, respectively (Fig. 5e, f). A large variation in the AHE response of the LAO/STO interfaces is observed, tuned by the associated B-site cation defect distribution. As shown in Fig. 5g and Supplementary Fig. 10, the hysteretic AHE part of the 5 K $R_{xy}$ for the B-site deficient sample was revealed after subtraction of LOHE contribution. There is also AHE signal in the B-site cation-rich LAO/STO around zero field (see Fig. 5f and Supplementary Fig. 9d), but with much smaller amplitude when compared with the B-site cation-deficient samples and limited to $T = 2$ K. This reflects the measured magnetic moment of the samples in SQUID magnetometry (Fig. 1c) and implies that the hysteresis in the Hall resistance likely originates from an additional contribution of magnetic moments induced by the B-site cation deficiency. Therefore, the AHE is expected to be proportional to the total magnetization of the system, $M_z$. Indeed, the coercive field ($B_C = {\sim}40{-}53$ mT) of the deconvoluted 5 K $R_{xy}$ (AHE) in the B-site cation-deficient LAO/STO is consistent with the coercive field ($H_C = {\sim}40{-}45$ mT) of the 5 K out-of-plane magnetic hysteresis, determined by the SQUID magnetometry (see Fig. 5i; details of deconvolution are

provided in Supplementary Note 4). These observations confirm that the high-mobility 2DEG can be effectively coupled with B-site cation defect-induced magnetism. Therefore, we assign the observed large enhancement in the AHE to a magnetic proximity effect between the conduction electrons of the 2DEG and the B-site cation defect-induced magnetic moments normal to the interface. These magnetotransport phenomena support the existence of a perpendicular magnetization across the LAO–STO interface, induced by B-site cation defect formation, and show a high tunability of the magnetoelectric properties of the interface by controlling the defect-induced magnetic structure.

## Discussion

Our findings show the formation of interfacial magnetism that stems from the atomic charge compensation across the interface region between the B-site cation-deficient LAO and STO. Such cation exchange/intermixing behaviours are commonly observed around complex oxide interfaces, which can be strongly dependent on the stoichiometry of oxide overlayers[23,24,44]. Our results indicate that the Al vacancy sites in the LAO overlayer play a crucial role in initiating the creation of secondary magnetic components and consecutive defect alignment. The formation of Ti vacancies on the STO side is sequentially related to the Ti substitute of (Ti$_{Al}$) from the surface of the STO to the Al-deficient LAO overlayer via preferential positioning of Al vacancies towards the interface (Fig. 3b, c, Supplementary Fig. 6). These cation defects induce large spin polarization at the neighbouring O atoms. However, in our work, it was found that the randomly distributed B-site cation vacancies,

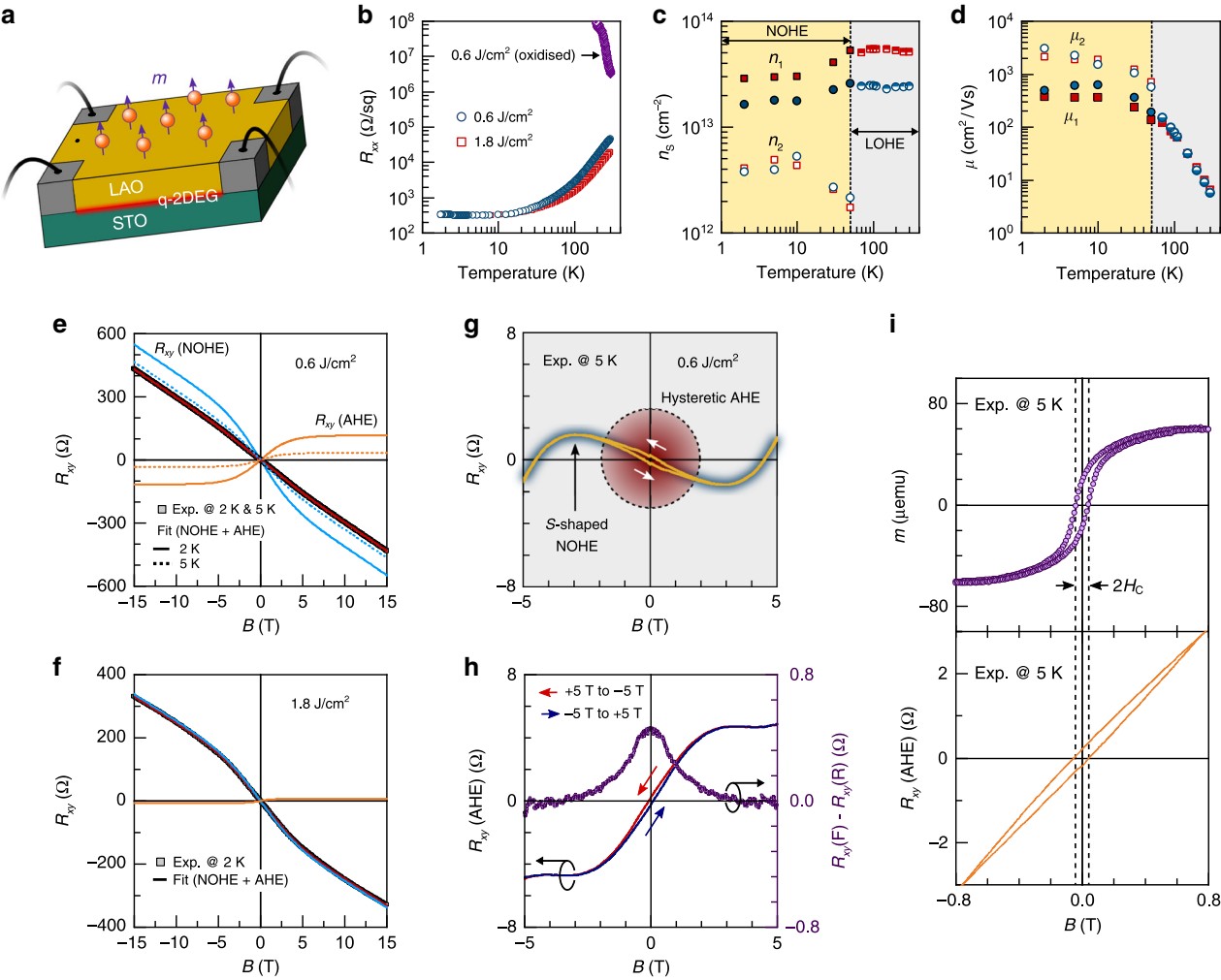

**Fig. 5 Anomalous Hall effect. a** A schematic showing the measurement of the B-site cation-deficient LAO/STO sample in a Van der Pauw geometry with spontaneous out-of-plane-ordered magnets. **b** Temperature-dependent sheet dc resistance ($R_{xx}$) of the B-site cation-rich ($E_g = 1.8$ J/cm$^2$, red squares) and B-site cation-deficient ($E_g = 0.6$ J/cm$^2$, dark blue dots) 9-u.c. LAO/STO systems, measured at $B = 0$ T. The fully oxidized sample ($E_g = 0.6$ J/cm$^2$, purple triangles) is highly resistive and insulating at low temperatures (< 100 K). **c, d** Sheet carrier density (**c**) and mobility (**d**) of the B-site cation-rich ($E_g = 1.8$ J/cm$^2$, red squares) and B-site cation-deficient ($E_g = 0.6$ J/cm$^2$, dark blue dots) 9-u.c. LAO/STO systems. The carrier transport of the samples was divided into two regimes, LOHE above 50 K and NOHE below 50 K. In the NOHE regime, the multiband conducting parameters [low-mobility ($\mu_1$) carrier density ($n_1$) and high-mobility ($\mu_2$) carrier density ($n_2$)] of the samples were derived using a two-band model. **e** A 2 K and 5 K Hall resistance, $R_{xy}$, of the B-site cation-deficient sample. The experimental 2 K–(5 K–) $R_{xy}$ is fitted by deconvoluting NOHE and AHE parts. **f** A 2 K $R_{xy}$ of the B-site cation-rich sample. **g** Hysteretic AHE and S-shaped $R_{xy}$ parts of the B-site cation-deficient LAO/STO, measured at 5 K and after subtracting the LOHE part ($R_{xy} = R_0B$) to the total $R_{xy}$ (Exp.). **h** The deconvoluted hysteretic AHE part and the difference of the forward $R_{xy}$ (F) and reverse $R_{xy}$ (R) while sweeping the fields at 5 K. **i** A 5 K out-of-plane magnetic hysteresis loop (in the upper panel) and the deconvoluted AHE part (in the lower panel) of the $R_{xy}$ for the sample. Dashed lines indicate the coercive field ($H_C = \sim45$ mT) of $m$ and $R_{xy}$ (AHE) while sweeping the applied fields.

which produce local p-type-like doping, can only sustain under sufficient oxidation conditions. Primarily, in a defective system, the negatively charged cation vacancies promote the formation of nearby compensating defects, $V_O$. Thereby, the oppositely charged defects energetically prefer to be the nearest neighbours by forming defect complexes to compensate for the charge of the BO$_2$ planes on both sides of LAO and STO. This also reveals that residual Al vacancies far from the interface (or the surface of LAO) can be eliminated by oxygen vacancy redistribution in the LAO overlayer, which corresponds to the observed magnetic saturation with the LAO thickness. Such charge compensation and/or interactions readily cause a large reduction in the cation vacancy-induced interface magnetization as observed from the conducting Al-deficient LAO/STO interface: a significant decrease in the magnetic moment of the Al-deficient LAO/STO sample occurs when cooled down in $P_g$ as an oxygen-deficient condition after film growth

(Supplementary Fig. 8). This clearly indicates that for the case of B-site cation-deficient LAO/STO systems the interfacial magnetism primarily originates from the cation defects in the interface rather than contributions by the distribution of $V_O$. In contrast, the electronic and magnetic nature of the Ti$_{Al}$$^{4+}$ anti-site defect in the sublattices of LAO differ entirely, when compared with B-site cation vacancies. Moreover, the total moment remains robust $\sim0.9\mu_B$ per defect pair with the equivalent positions of $V_O$. Given that the two consecutive Ti$_{Al}$ and $V_{Ti}$ cation defects are preserved at the interface as stable dipoles, this is the origin of the magnetic properties at the interface. Evidently, it was found that the cation defect-induced magnetization of the B-site cation-deficient interface remains robust with varying $V_O$ distribution.

Despite the effect of $V_O$ on the degradation of the defect-associated interfacial magnetism, our observations suggest that the coexistence of spontaneous cation defect ordering and $V_O$ in

the buried interface is required for the emergence of both the interfacial conductivity and magnetism. Furthermore, the B-site cation defect alignments locally distributed near the interface are coincidently accompanied by diminishing the induced built-in field of the LAO layer. Although this coupling partially hinders the resultant lower charge confinement at the interface, it is still sufficient to drive the structural distortion and orbital reconstruction of the STO surface (Supplementary Note 3 and Supplementary Fig. 7a). This diverges deeper into the STO bulk, wherein reside the multiband charge carriers. The inclusion of oxygen vacancies as a source of conduction electron into the generated 2D Ti $d$-bands at the interface ties up the observed multiband charge carrier distribution[15,44]. At low temperatures ($\leq 50$ K), the motion of the confined $d_{xy}$ electrons couples with the preferential out-of-plane magnetic moments of the B-site cation defects, resulting from a magnetic proximity effect. Hence, a complementary control of the B-site cation deficiency and electrostatic boundary condition of the upper polar layer is essential to restore the observed emergent electronic and magnetic structures at the strongly correlated heterointerface to trigger the effective magnetoelectric coupling effect. Such coupling effects at the interface can be effectively driven by controlling the B-site cation defect distribution without employing foreign magnetic elements and layers. In fact, the evoked out-of-plane magnetic moments, induced by the defect ordering, have enormous importance on both the magnetism and electronic properties of the system because the properties can be highly tunable by an external stimulus (e.g., the electric control of charge and spin states) for potential spin-based energy applications with high density, high stability, and low power consumption.

In this work, we show direct evidence that B-site cation defects are the main source for the magnetic ordering at the complex oxide interface. Cation defects are thus an important additional knob to tune magnetism in oxide heterostructures. Especially, in the case of LAO/STO interfaces, it is a key concept to create and modify the consecutive cation defect-associated magnetism. Our experimental and theoretical results reveal the significant compensating behaviour of oxygen vacancies to the induced magnetic reservoirs at the interface—the charge acceptor-like B-site cation vacancy defects (both Al and Ti vacancies). The spontaneous defect assembly causes the emergence of perpendicular magnetic clusters/ordering at the B-site cation-deficient interfaces through the atomic charge compensation of the B-site cation vacancy defects. This tunability of the interface magnetism offers a great step forward exploring and engineering these exciting rich oxide interface materials.

## Methods

**Film preparation**. Epitaxial LAO ultrathin film layers were grown on $TiO_2$-terminated STO(001) substrates [$5 \times 5 \times 0.5$ ($z$) mm$^3$ size] by using PLD (a KrF excimer laser, $\lambda = 248$ nm), equipped with an RHEED. To adjust the film stoichiometry, all of the films were grown by various laser energy fluence from ~0.6 to ~2.0 J/cm$^2$ with a repetition rate of 1 Hz. The growth temperature and $P_g$ were 700 °C and $5 \times 10^{-4}$ mbar, respectively. The oxygen vacancy content in the grown samples were controlled through oxygen cooling process: the samples with higher $V_O$ contents were made by cooling in the same $P_g$ after the sample growth, while the oxidized samples were made by cooling in a high oxygen atmosphere ($P_O = 200$ mbar).

**Characterizations**. The surface morphologies of the grown films were determined by RHEED patterns and atomic force microscopy. XPS measurements were performed under ultra-high vacuum (base pressure $= 3 \times 10^{-11}$ mbar) using an Omicron SPHERA hemispherical analyser and a monochromatic Al K$\alpha$ X-ray source ($hv = 1486.6$ eV). The emitted photoelectrons from the surface of the samples were collected with variable TOAs ($\theta = 30°$ and 90°). During the XPS measurements, surface charging effects were compensated by using a low-energy electron flood gun (Omicron CN10). The binding energy scale was calibrated to the C 1$s$ core-level peak at the binding energy of 284.5 eV with an overall energy resolution of 470 meV. The XPS spectra were fitted using a Shirley background

subtraction and Voigt line shapes. Compositional ratios of the elements (La, Al, Sr and Ti) in the films were determined after correcting for the IMFP (inelastic mean free paths) of the photo-emitted electrons via the TPP-2M formula[45] and by applying a Scofield photoionized cross-section[46]. The maximum probing depths of the surface-sensitive XPS for Al 2$p$, La 4$d$, Sr 3$p$ and Ti 2$p$ core levels of LAO and STO, collected with a 30° TOA, were estimated to be ~4.7, ~5.6, ~5.6 and ~3.3 nm, respectively. Magnetic moments were measured in a Quantum Design SQUID Magnetometer with samples mounted in different orientations to measure along the in-plane and out-of-plane crystallographic directions. The XMCD experiments were performed at the BL29 BOREAS beamline at the ALBA synchrotron radiation facility. The X-ray absorption was measured using linear polarized light with the photon with respect to the sample geometry ($I//c$ and $I//a$), and circular polarized light with the photon spin parallel ($\rho^+$) or antiparallel ($\rho^-$) with respect to the magnetic field. The spectra were collected with the beam on a tilted incidence (20° to the film surface) and in normal incidence. The spectra were recorded using the total electron yield method (by measuring the sample drain current) in a chamber with a vacuum base pressure of $2 \times 10^{-10}$ mbar. For the electrical measurements, the measured sample size is $5 \times 5$ mm$^2$ and contracts are achieved by Ohmic Al wire bonding on all the edge corners of the sample. The magnetotransport measurements were carried out in a CRYOGENIC cryogen-free measurement system with the temperature ranging from RT to 2 K and perpendicular magnetic fields up to $\pm 15$ T. To provide additional detailed information on the determination of the AHE in samples, we performed Hall measurements in a Hall bar-type configuration using the rectangular-shaped samples, that is, charge carriers flow parallel to the single longitudinal channel and the transverse Hall voltages of the opposite sides of the channel are measured. The magnetic fields were applied perpendicular to the $xy$-plane of the samples and the Hall resistance was measured in a five-loop sequence of the magnetic field (e.g., $0 \rightarrow +7$ T to $-7$ T $\rightarrow +7$ T to 0 T) with refined field intervals (1–2 mT/s). Both the high-field ($B \geq 13$ T) and low-field ($B \leq 7$ T) data were used for the two-band and AHE analyses, respectively. Coercive fields, $B_C$, of the extracted hysteretic 5 K $R_{xy}$ (AHE) parts were determined by $B_C = (|+B_C| + |-B_C|)/2$, where the values of $+B_C$ and $-B_C$ are $+45 \pm 5$ and $-49 \pm 4$ mT, respectively. Each $B_C$ was obtained from a zero-crossing field, where the $R_{xy}$ (AHE)–$B$ hysteresis loop crosses zero $R_{xy}$ during the field apply. TEM lamella were prepared by focused ion beam and imaged in STEM mode on a double spherical aberration (Cs)-corrected Thermo Fisher Scientific Titan Themis 60–300 operated at 300 kV. For the high-angle annular dark-field STEM images, a camera length of 115 mm and a beam current of 90 pA were employed. The microscope was equipped with a Gatan GIF Quantum spectrometer and EELS was performed using the Gatan high-angle annular dark-field STEM detector. The spectrum imaging was carried out in dual EELS mode and all the spectra were acquired at 0.05 eV per channel dispersion in an energy resolution of 0.9 eV.

**Crystalline, electronic and magnetic structure calculations**. First-principles calculations were carried out within the DFT using the projector augmented-wave method as implemented in the Vienna ab initio simulation package (VASP) code. The exchange-correlation energy was treated using the generalized gradient approximation. The energy cutoff for the plane-wave expansion was set to 450 eV. All structural optimizations were obtained with a conjugate-gradient algorithm and a force tolerance criterion for convergence of 0.01 eV/Å. The Brillouin zone integration was performed using $4 \times 4 \times 4$ $k$-point Monkhorst–Pack mesh. The electronic and magnetic structures were also investigated with a self-consistent Green function method specially designed for surfaces and interfaces. For these calculations, we used as input the crystalline structure obtained from the VASP structure optimizations.

## Data availability

The authors declare that the data supporting the finding of this work are available within this article and its Supplementary information files, and are available from the corresponding author on reasonable request.

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

## Acknowledgements

We thank L. Navratilova, S.C. Sandu and M. Cantoni at the Interdisciplinary Centre for Electron Microscopy-EPFL for their technical support in the preparation of the TEM samples. We also thank D. Eilmsteiner and S. Paischer in the Institute of Theoretical Physics, Johannes Kepler University and L. Chotorlishvili in the Institut für Physik, Martin-Luther-Universität Halle-Wittenberg for their theoretical support and discussion. D.-S.P., A.D.R., I.V.M., S.O., A.B., I.M., K.D. and A.E. acknowledge the financial support from the Deutsche Forschungsgemeinschaft through the Collaborative Research Centre *Sonderforschungsbereich* 762. This work was partially supported by the European Commission by grant 80127 (project Biowings H2020 Fetopen 2018-2022). A.K. and G.S. acknowledge the financial support by the Swedish Research Council (Vetenskapsrådet) and Swedish infrastructure for micro- and nanofabrication (MyFab).

## Author contributions

D.-S.P. conceived the idea, designed this research project and fabricated thin film samples. A.D.R., D.-S.P., Z.L., G.J.R. and K.D. conducted and interpreted the magnetic measurements. Y.G., Y.C., G.S., D.-S.P., A.K. and N.P. performed the transport and magnetoresistance measurements and analyses. I.V.M., S.O., I.M. and A.E. carried out and analysed the DFT calculations. M.W. and D.-S.P. completed and interpreted the XPS measurements. J.L., P.M. and V.T. performed the STEM measurements and analyses. S.A. and J.H.-M. carried out the XAS measurements in the ALBA synchrotron light source (beamline BL29 BOREAS) in Barcelona, Spain. All authors discussed the results. The paper was written by D.-S.P. and N.P. with assistance from G.J.R., I.M., A.B. and A.E.

## Competing interests

The authors declare no competing interests.
