## [Peer Review File · Nature Communications]

Reviewers' comments:

Reviewer #1 (Remarks to the Author):

In their manuscript, the authors make a central claim that B-site cation stoichiometry is crucial for creation and control of magnetism at the LAO/STO interface.

DFT calculations show magnetic moments on the B site cation vacancy, consistent with EELS, magnetometry. The story is pretty convincing, and overall this part of the paper seems like a significant advance.

However, the second part of the paper that tries to connect anomalous and nonlinear Hall effect measurements is much less convincing. A basic question arises: if there is ferromagnetism, why is there no hysteresis in the anomalous Hall effect? This discrepancy is not answered. Instead, there is a lot of repeated discussion of nonlinear/AHE that are well documented in the literature.

I feel that there are really two distinct sets of results reported in this paper. I would support in principle publication of a revised manuscript that sticks to the cation vacancy data and analysis. The second part of the manuscript I do not feel rises to the same level of significance or clarity.

I have a few other comments below that should be addressed:

In Figure 1, a diamagnetic/paramagnetic background is subtracted. The origin of this diamagnetic background is not discussed or explained. It too seems to depend on the growth conditions, and is different for the various laser fluencies explored. It is not clear why this is referred to as a "correction". It is a measured response that was also reported by Ariando (Ref. 10).

Only a ~3% difference in La and Al atomic ratio is observed at higher laser fluence which goes to a maximum of 10% at lower laser fluencies. The authors claim the samples to be La rich at low laser fluencies and Al rich at high laser fluencies. What is the base line with respect to which the samples are considered to be Al rich or La rich?

Also, why is the magnetism later assumed to be predominantly arising due to Al deficiency and not La richness?

In Figure 2a. the authors observed hysteresis loops in 9uc LAO/LAO (001) to be 21% smaller compared to 9uc LAO/STO sample. The authors suggest that this implies that a substantial part of the total moment measured in the Al-deficient LAO/STO system arises from the underlying STO layers. However, the Sr and Ti atomic ratio almost remains unchanged according to Figure 1b. How should the reader interpret the relation between the atomic ratios and the induced magnetism?

In addition to the 9uc LAO/LAO samples, strong hysteresis loop is observed in 9uc STO/STO (001) samples, which is dependent on laser fluencies. The results observed is comparable to that of 9uc LAO/STO at mid-range laser fluencies. Therefore, how is the observed magnetism in the case of 9uc LAO/STO sample attributed to as a property of the LAO/STO interface, due to defects and atomic compensation?

Reviewer #2 (Remarks to the Author):

The authors have investigated the magnetic and magnetotransport properties of LAO/STO and STO/STO film/substrates grown using different energy fluences with PLD. The authors claim that variation of B-site occupation affects the magnetism in p-doped films. Specifically, the authors suggest the magnetism comes from a combination of spin-polarization of oxygen (which lacks experimental proof) and a moment on Ti (which is seen in the XMCD). For the reasons discussed below I am not convinced and cannot recommend publication of the manuscript as is.

1)

Several of the figures show the magnetism as moment, which is labelled as capital M, in units of micro emu, rather than magnetization. This produces a couple of problems.

First, have the moments in Fig 1a been normalized to sample size for all the references? Obviously, the more sample the more measured moment.

The second problem is the most alarming to me. Take Fig. 1c—the moment saturates at 0.6 J/cm^2 . Here the moment is $\sim 50 \times 10^{-6} \text{ emu}$. From the methods section, the sample's area is $5 \text{ mm} \times 5 \text{ mm}$. From Fig 3d, there are Ti ions (these produce some of the moment the authors claim) within a region of the LAO $\sim 2 \text{ nm}$ thick. These numbers yield a volume of magnetic material $\sim 5 \times 10^{-8} \text{ cm}^3$.

Ratioing the moment to the volume I get a magnetization of 1000 emu/cc ($\text{emu/cc} = \text{kA/m}$). The magnetization of iron metal is 1740 kA/m . There is something very wrong.

Which direction is the field applied?

2)

Fig. 1c shows a diminishment of the moment with energy. This correlation is compelling. What I do not find compelling is the assertion that the Al and Ti concentration (and hence the related vacancies) change with energy. To me the Ti concentration appears constant in Fig 1b. In the case of Al, the claim that the concentration changes is based on one data point at 1.8 J/cm^2 . Take that one point away and the entire thesis disappears.

3)

The authors say the induced moment on Ti is important to the overall moment. Fig. 2a shows data for a moment in a sample that doesn't have any Ti in it. Is the difference between the two curves due to the induced Ti moment, and the rest is spin-polarization on O? Why wasn't XMCD of O measured, as done by others?

4)

Fig. 2d shows that the moment decreases with increasing energy in STO film/STO substrate. The magnitude of the moment is $3/5$ of the LAO film/STO, so nearly the same. The authors suggest the polar vs. not polar character of LAO/STO induces a reconstruction that promotes O spin polarization and Ti redistribution and its induced moment. In the case of STO/STO this mechanism is lacking, and yet the magnetism is nearly as strong as the LAO/STO case. I sense an inconsistency. Further the lack change of Ti concentration with energy (see Fig. 1b) suggests to me the STO/STO sample is fairly uniform, so what is special about the STO film, when the STO substrate does not show magnetism? The authors say there are no impurities, yet what is the evidence for this claim? In (1) I noted the magnetization seems to be non-sensical. If there is some error related to (1), then what concentration of Fe impurities is needed to produce the magnetism? I propose an alternate hypothesis: Magnetic impurities are deposited into the films, e.g., concomitant bombardment of the chamber, and the amount of impurities increases with energy of the PLD.

5)

I would say "atomic charge compensation" not "atomic compensation".

Reviewer #3 (Remarks to the Author):

The manuscript 'The emergence of magnetic ordering at complex oxide interfaces tuned by defects' by Park et al. reports on magnetic properties of LAO/STO heterostructures produced with varying LAO cation stoichiometry. The authors suggest a mechanism based on the formation of B-site cation defects, namely, titanium vacancies on the STO side, and TiAl antisite defects on the LAO side of Al-deficient LAO layers to be the main driver of magnetism in the system.

The magnetism in these systems is clearly a very interesting and impactful topic to be addressed in the field. Experimental and theoretical work as well as data evaluation is appropriate and the presented work can be reproduced based on the given information. The authors present a number of intriguing experimental data, particularly, systematic stoichiometry-dependence of the magnetic signal, and support their conclusions by DFT calculations.

While I like the idea of the paper and while I am impressed by the reported data, I am more reluctant about the details in the presented argumentation. I am missing a couple of points that should be addressed in order to arrive at a full understanding of the data. Therefore, I can recommend publication of the manuscript only after careful revision.

My main concerns are the following:

1. The authors argue that the magnetism at the LAO/STO interface is p-type in nature and argue that the XMCD data obtained for the O K-edge gives evidence for this. The signal however is very small and the discussion of the data in the SI is not detailed enough. In order to be convincing, the authors should provide field-dependent data, indicating that the XMCD is robust and behaving systematic with field (or at least a field-swap). If confirmed, the discussion should be moved to the main paper – as current state of knowledge (see e.g. Refs. 16,17) is that the interface shows dxy-type magnetism. It is also not clear to me how the XMCD (Fig. S7) can 'verify the magnetization of the anticipated anti-site defects' (statement on page 5) as it is lacking the sensitivity to single defect species?

2. The authors should address in more detail, why there is a discrepancy between transport (AHE arises at low temperature) and SQUID data (which indicates strong magnetism up to RT). As the authors argue that the magnetism is very robust also at room temperature, why is there no anomalous Hall effect observed at room temperature?

3. The authors do not address the dynamics of the proposed defect formation process – the diffusion of B-site cations in perovskites is typically very slow hindering the formation of B-site vacancy defects (and often favoring A-site defect formation). Also, the formation energy for B-site vacancies is typically large (e.g. Akhtar, Catlow, J. Am. Ceram. Soc. 78, 421 (1995))? It would be important the authors address in more detail how is the Ti moving into the LAO layer to form anti-site defects? Also the charge-state of the TiAl anti-site defect should be discussed – where is the extra electron in the LAO layer going?

4. I think it is very interesting finding that homoepitaxial STO and LAO can both show magnetism when grown in a non-stoichiometric manner, which nicely scales with layer thickness. It is not clear for me (this may be a matter of clarity only), why for LAO/STO the magnetization saturated, as I would always expect to see the thickness-dependent magnetization of the thin film on top of the one at the interface? The authors argue that in the heterostructure parts of the cationic defects are co-compensated by oxygen vacancies. This is a straight forward assumption, but why is it not happening in the homoepitaxial thin films, which following this arguments should not show magnetization?

5. The authors rule out other defect scenarios (oxygen vacancies and Sr vacancies) based on DFT, but they do not show their data (page 4, bottom). This should be done at least in the SI as both

statements are surprising – oxygen vacancies have been identified in the literature to be a possible origin of magnetism in STO (see e.g. Refs. 17/18). For Sr vacancies one may expect a similar p-type hole formation as observed in the Ti vacancy case – it is hence interesting to understand in how far they result in different behavior?

6. A quantification for the magnetic data should be given. Based on SQUID, one should estimate the number of anti-site defects at the interface.

Besides these major comments, I have some minor comments the authors should consider:

- On page 3, I would recommend to mention the growth pressure in the main text, as it is important for the required plasma-gas interaction responsible for the stoichiometry variation.
- Page 3: The term 'angle-dependent' XPS is misleading as measurements were made only at one angle. I suggest to remove. Also, the information depth should be mentioned for the individual core levels. As the authors argue in the SI, the information depth will be different for each element/core level.
- Page 3: 'reduction of Al' – I would suggest to use 'reduction of concentration of Al' to avoid confusion with a valence-change process.
- Fig. 1c – can the authors comment on why the substrate gives a paramagnetic signal, while for the LAO/STO samples a diamagnetic background is observed?
- The section on transport analysis and the data treatment is mainly following Ref. 40, which should be indicated more clearly in the main text. Also, a reference to Joshua, Nat. Comms. 3, 1129 (2012) should be added.
- Fig. 3h – data seems to be not consistent to Fig. 2a/3a? Why is the saturation magnetization in perpendicular field much bigger than anticipated in 2a/3a?
- Fig. 4 – it is very interesting that the non-linearity of normal Hall effect and AHE almost cancel out, resulting in an almost linear R_{xy} . Can the authors comment how stable the fitting of the data is in this case?
- Page 6: It is not clear to me, why a proximity effect of the hysteretic out-of-plane magnetization in the LAO layer on the 2DEG does not yield a hysteresis in the observed AHE?
- XPS survey scans in SI – there are two core levels labeled with La 3d.
- XAS O K edge: the authors should indicate also positions of Sr hybrids for clarity.

Reply to Reviewers:

Reviewer #1:

In their manuscript, the authors make a central claim that B-site cation stoichiometry is crucial for creation and control of magnetism at the LAO/STO interface.

DFT calculations show magnetic moments on the B site cation vacancy, consistent with EELS, magnetometry. The story is pretty convincing, and overall this part of the paper seems like a significant advance.

However, the second part of the paper that tries to connect anomalous and nonlinear Hall effect measurements is much less convincing. A basic question arises: if there is ferromagnetism, why is there no hysteresis in the anomalous Hall effect? This discrepancy is not answered. Instead, there is a lot of repeated discussion of nonlinear/AHE that are well documented in the literature.

Reply to comment: We appreciate the comments of the Referee #1 for his/her positive feedback regarding our central claim that B-site cation stoichiometry is crucial for creation and control of magnetism at the LAO/STO interfaces. Following the comment, we have conducted additional low-temperature magnetoresistance measurements to address for the hysteretic AHE in a B-site cation deficient LAO/STO sample. Also, we have revised and simplified our manuscript accordingly.

We agree with the Referee's comment about the need to show the hysteretic anomalous Hall effect (AHE) in the system. Indeed, in our previous results, the hysteretic AHE was not clearly observed near the zero-field (field intervals of 20 – 125 mT/sec). We have therefore performed additional magnetotransport measurements of a B-site cation deficient LAO/STO sample (grown by $E_g = 0.6 \text{ J/cm}^2$) with much fine field intervals (field steps: 1 - 2 mT/sec). These additional results show a clear hysteretic behaviour in the AHE. The coercive strength (40 – 50 mT) of the hysteretic AHE is in line with the H_C (40 -45 mT) of the ferromagnetic B-site cation deficient LAO/STO sample as shown below in Fig. S10. This reveals that the weak magnetism at the 2DEG interface can be coherently coupled with, and enhanced by the B-site cation defect-induced magnetism. To support our findings, the new results and statements have been added in the revised manuscript and supplementary information with a new Figure S10.

- In the revised manuscript, we have added new statements as:

“Furthermore, a hysteretic AHE feature in the B-site cation-deficient LAO/STO sample was observed near zero-field while sweeping the fields (see Fig. S10). The coercive field ($B = \sim 40 - 50 \text{ mT}$) of the deconvoluted R_{xy} (AHE) in the B-site cation deficient LAO/STO is consistent with the magnetic coercive field ($H_C = \sim 40 - 45 \text{ mT}$), determined by SQUID magnetometry. These indicate a coupling effect between the defect-induced magnetic moments and the 2DEG. In addition, the hysteretic AHE of the 2DEG visibly appears up to 5 K. This is in contrast to a strong temperature-dependent hysteretic AHE feature in conventional 2DEG interfaces, usually limited by very low temperatures ($T < 2 \text{ K}$) in previous reports⁴⁰⁻⁴³. These observations further confirm that the weak magnetism of the 2DEG reported until now can be effectively coupled and enhanced by much stronger B-site cation defect-induced magnetism.”

- In the supplementary Information, we have added new statements:

“To clarify the coupling effects between the defect-induced magnetic moments and the 2DEG near zero field, magnetotransport measurements of a B-site cation-deficient LAO/STO sample were carried out with refine field intervals (1 – 2 mT/sec) considering the measurement time (/field interval) dependence of the ferromagnetic coercive strength of the system. As seen in Figs. S10b,e, a clear hysteretic R_{xy} appears at the zero field while sweeping the fields. This corresponds to an increase in the Hall coefficient, R_H , towards zero field and a spiky negative R_H then occurs when the applied field is reversely switched across the zero field due to the magnetic remanence of the system (Fig. S10f). The coercive field ($B = \sim 40 - 50$ mT) of the deconvoluted R_{xy} (AHE) part of the B-site cation deficient LAO/STO is consistent with the magnetic coercive field ($H_C = \sim 40 - 45$ mT), determined by SQUID magnetometry (Fig. S10g). In addition, the hysteretic AHE of the 2DEG visibly appears up to 5 K. These observations further confirm that the weak magnetism of the 2DEG reported until now can be effectively coupled and enhanced by much stronger defect-induced magnetism.”

Figure S10. (a) Linear ordinary Hall effect (LOHE) and anomalous Hall effect (AHE) fitting parts of the experimental 5 K-Hall resistance [R_{xy} (Exp.)] of a B-site cation deficient LAO/STO sample, grown with $E_g = \sim 0.6$ J/cm², measured in a magnetic field of 5 T with a field sweep of 1 mT/sec. (b) The remanent R_{xy} (Exp.) of the sample at the zero field while sweeping the fields. (c) Residual S-shaped R_{xy} over the applied field and hysteretic AHE in small field range in the deconvoluted AHE part after subtracting the LOHE part ($R_{xy} = R_0B$) to the total R_{xy} (Exp.). (d) Nonlinear ordinary Hall effect (NOHE) and AHE fitting parts of the R_{xy} (Exp.) of the sample. (e) The deconvoluted hysteretic AHE part and the difference of the forward and reverse R_{xy} (Exp.) while sweeping the fields at 5 K. In a closed field loop, the R_{xy} (F) was measured in a forward field sweep (+5 T to -5 T) and the R_{xy} (R) was measured in a reverse field sweep (-5 T to +5 T). (f) The hysteretic AHE part of the R_H (Exp.) of the sample, measured in the field range of 5 T. The hysteretic AHE part visibly appears in the applied field below 1 T. Correspondingly, a spiky negative R_H occurs while the applied fields cross over the zero field and the negative R_H remains up to $B \sim 40 - 50$ mT due to the magnetic remanence and the corresponding remanent R_{xy} at the zero-field. (g) A 5 K out-of-plane magnetic hysteresis loop (in the upper panel) and the R_{xy} (AHE) (in the lower panel).

deconvoluted AHE part (in the lower panel) of the R_{xy} of the sample. Dashed lines indicate the coercive field ($H_C \sim 45$ mT) of the magnetic moment and R_{xy} while sweeping the applied fields.

I feel that there are really two distinct sets of results reported in this paper. I would support in principle publication of a revised manuscript that sticks to the cation vacancy data and analysis. The second part of the manuscript I do not feel rises to the same level of significance or clarity.

Reply to comment: We thank the reviewer for his/her comments and we therefore tried to make our arguments clearer. Our findings show that interfacial magnetism in LAO/STO system can be effectively tuned by the distribution of B-site cation defects. We have further showed that the defect-induced interfacial magnetism can be significantly coupled with the interface conductivity, primarily created by the presence of oxygen vacancies. Hence, our results provide clear evidence and methodology of designing defect assembly and tuning the interfacial magnetism.

I have a few other comments below that should be addressed:

In Figure 1, a diamagnetic/paramagnetic background is subtracted. The origin of this diamagnetic background is not discussed or explained. It too seems to depend on the growth conditions, and is different for the various laser fluences explored. It is not clear why this is referred to as a “correction”. It is a measured response that was also reported by Ariando (Ref. 10).

Reply to comment: To eliminate any external artifacts on the magnetic signals of the SUQID magnetometer samples, we conducted preliminary background measurements without any sample. Any diamagnetic and paramagnetic background components of the samples are therefore mainly due to instrumental accessories (such as the sample holder), see Fig. R1. This is why we used the word “corrected”, when describing the procedure in the manuscript. To further clarify this, we added this new data into the revised supplementary information (S3) as follows:

Figure R 1. (a) 5 K magnetic hysteresis loop of the sample holder used in the SQUID magnetometer, applying fields of +5 and -5 T. A paramagnetic (PM) background component was defined by subtracting the negative slope of the diamagnetic (DM) component. (b) 5 K magnetic hysteresis loop of an annealed TiO₂-terminated STO substrate, mounted on the SQUID sample holder by low-temperature glue. (c) A hysteresis loop in the range of + and -0.3 T for the STO substrate. After subtracting the DM and PM background components, a real magnetic hysteresis loop of the STO substrate which shows no ferromagnetism is observed with a moment of $\sim 1 - 2 \times 10^{-7}$ emu, which is not comparable with the measured moments of the LAO/STO samples.

Only a ~3% difference in La and Al atomic ratio is observed at higher laser fluence which goes to a maximum of 10% at lower laser fluencies. The authors claim the samples to be La rich at low laser fluencies and Al rich at high laser fluencies. What is the base line with respect to which the samples are considered to be Al rich or La rich?

Reply to comment: We thank the reviewer for rising this question. In our work, we have taken the cation-stoichiometric ABO₃ perovskite (50 % A-site and 50 % B-site cations) base-line. To clarify this, we have added the following statement in the revised manuscript:

“Cation stoichiometry of ABO₃ perovskite with 50 % A- and 50 % B-site cations is taken as the base-line of the cation-stoichiometry in the grown films.”

Also, why is the magnetism later assumed to be predominantly arising due to Al deficiency and not La richness?

Reply to comment: In ABO₃ perovskites, the formation of La interstitials in LaAlO₃ is typically energetically unfavourable with respect to the ionic radius of the constituent elements and their thermodynamics, other secondary phases such as superstructures could be formed breaking the AO-and-BO₂ plane-altered perovskite cell structures with a higher off-stoichiometry. Also, from our experimental work (atomic resolution-STEM) and theoretical calculations, the B-site cation deficient LaAlO₃ upper layer clearly sustains an ABO₃-type perovskite structure. The Al-vacancy is the main reason for the interfacial magnetism in this system. Therefore, we show that Al-deficiency of the LaAlO₃ film composition creates the interfacial magnetism.

In Figure 2a. the authors observed hysteresis loops in 9uc LAO/LAO (001) to be 21% smaller compared to 9uc LAO/STO sample. The authors suggest that this implies that a substantial part of the total moment measured in the Al-deficient LAO/STO system arises from the underlying STO layers. However, the Sr and Ti atomic ratio almost remains unchanged according to Figure 1b. How should the reader interpret the relation between the atomic ratios and the induced magnetism?

Reply to comment: We appreciate the comment of the reviewer and to avoid any misunderstanding, we corrected the text in accordance with reviewer's comments:

“As observed in Fig. 2a, at 5 K the saturation of the magnetic moment of the homoepitaxial sample (9uc-LAO/LAO(001)) is approximately 21 % lower than 9uc-LAO grown on a STO substrate.”

To

“As observed in Fig. 2a, at 5 K the saturation of the magnetic moment of the homoepitaxial sample (9uc-LAO/LAO(001)) is approximately 21 % of the moment of the Al-deficient 9uc-LAO/STO heterostructure.”

For the second question, the change in the Ti/Sr ratios in the LaAlO₃ films seems to be small when compared to that of the Al/La (Fig. 1b). However, when the Al deficiency is considered, there is indeed a significant change in the Ti/Sr as a function of the stoichiometricity of the LaAlO₃ layer. ~3.6 % Ti atomic ratio can compensate approximately 22.2 % of the 16.2 %-Al deficiency in the LaAlO₃ layer of the sample, grown by a laser fluence of 0.6 J/cm².

Growth of the LaAlO₃ upper layer with an Al-deficient composition initiates the creation of Al vacancies (V_{Al}) near the interface, as confirmed by DFT. This subsequently leads to the formation of substitutional Ti (Ti_{Al} antisite defects) in the Al-deficient LAO layer and Ti vacancies (V_{Ti}) in the STO.

In addition to the 9uc LAO/LAO samples, strong hysteresis loop is observed in 9uc STO/STO (001) samples, which is dependent on laser fluencies. The results observed is comparable to that of 9uc LAO/STO at mid-range laser fluencies. Therefore, how is the observed magnetism in the case of 9uc LAO/STO sample attributed to as a property of the LAO/STO interface, due to defects and atomic compensation?

Reply to comment: For the case of oxidized B-site cation deficient LAO/STO sample, grown by a laser energy fluence of 0.6 J/cm², a Ti atomic ratio of ~3.6 % is found in the B-site cation stoichiometry of the LAO film layer with ~16.2 % Al deficiency (determined by XPS), which can compensate approximately 22.2 % of the Al deficiency. Our experimental results and DFT calculations show that individual cation defects, both V_{Al} and Ti_{Al} in LAO and a V_{Ti} in STO, separately exist and can induce local moments, 0.87 μ_B/V_{Al}, 0.9 μ_B/Ti_{Al}, and 0.48 μ_B/V_{Ti}, respectively. Assuming the same amount (~3.6 %) of B-site cation vacancies in the STO giving ~3.6 % Ti out-diffusion to the LAO upper layer, 46.0 % (~22 μemu), 37.5 % (~18 μemu), and 16.5 % (~8 μemu) moments of the LAO/STO system can be separately addressed from V_{Al} and Ti_{Al} in LAO, and V_{Ti} in STO. This could reflect the discrepancy in the total magnetic moments of the heteroepitaxial LAO/STO and homoepitaxial LAO/LAO samples, presented in Fig. 2a in the main manuscript.

We agree with the Reviewer's comment that the measured moments of the homoepitaxial STO(001)/STO(001) samples [as well as LAO(001)/LAO(001) sample] cannot be quantitatively compared with the LAO/STO case due to the different kinetics of the arriving species during PLD deposition and the resultant defect density - this could be interesting subject for a future work. Moreover, the measured moments could originate mainly from B-site cation vacancy defects because of the same charge valence of any compensating ions in the STO lattices. However, it is clear that the variation in

the magnetic moments of the STO films strongly depend on the laser-energy-fluence during film growth, very similar to the case of heteroepitaxial LAO/STO. This provides direct evidence into the creation of ferromagnetic moments, driven by B-site cation deficiency in STO. So, this convincingly supports that the total magnetic moment of the B-site cation deficient LAO/STO system should be combined with any induced moments in the STO side as a consequence of Ti redistribution.

Reviewer #2 (Remarks to the Author):

The authors have investigated the magnetic and magnetotransport properties of LAO/STO and STO/STO film/substrates grown using different energy fluences with PLD. The authors claim that variation of B-site occupation affects the magnetism in p-doped films. Specifically, the authors suggest the magnetism comes from a combination of spin-polarization of oxygen (which lacks experimental proof) and a moment on Ti (which is seen in the XMCD). For the reasons discussed below I am not convinced and cannot recommend publication of the manuscript as is.

Reply to comment: We appreciate the comments of Reviewer #2 about our findings and we have tried to comply with his/her comments point-by-point in the revised manuscript which provide further information to support our observations. We do believe that his/her comments have helped for improving our manuscript.

1) Several of the figures show the magnetism as moment, which is labelled as capital M, in units of micro emu, rather than magnetization. This produces a couple of problems. First, have the moments in Fig 1a been normalized to sample size for all the references? Obviously, the more sample the more measured moment.

Reply to comment: We thank the reviewer for his/her comments. We have now corrected the symbol of the moment units to the conventional unit, “*m*”, in the revised manuscript to avoid any confusion for readers.

All the samples for the presented references in Fig. 1a [except samples in UNSW (Ref. 26 & 27), no information on the area/volume of STO substrate] are $\sim 5 \times 5 \text{ mm}^2$ size and the LAO layer thicknesses are highlighted showing no dependence of layer thickness, but it provides clear evidence that there is a strong PLD-growth-parameter dependence of the interfacial magnetism in the LAO/STO system. To clarify this, we added the following statement in the revised manuscript:

“The observed trend shows that relatively weak interfacial magnetism ($m \leq \sim 2 \mu\text{emu}$) commonly occurs with a low oxygen-pressure ($P_g \leq 10^{-4} \text{ mbar}$) growth atmosphere and a high-laser-fluence ($E_g > 1.2 \text{ J/cm}^2$) film growth, independent of the film thickness and size of the grown samples.”

The second problem is the most alarming to me. Take Fig. 1c—the moment saturates at 0.6 J/cm^2 . Here the moment is $\sim 50 \times 10^{-6} \text{ emu}$. From the methods section, the sample’s area is $5 \text{ mm} \times 5 \text{ mm}$. From Fig 3d, there are Ti ions (these produce some of the moment the authors claim) within a region of the LAO $\sim 2 \text{ nm}$ thick. These numbers yield a volume of magnetic material $\sim 5 \times 10^{-8} \text{ cm}^3$. Ratioing the moment to the volume I get a magnetization of 1000 emu/cc ($\text{emu/cc} = \text{kA/m}$). The magnetization of iron metal is 1740 kA/m . There is something very wrong.

Which direction is the field applied?

Reply to comment: Our work demonstrates that the total magnetic moment of the B-site cation deficient LAO/STO system is not solely from the LAO upper layer. It is due to a combined effect and is a result of atomic and/or charge compensating effect across the interface. So, we cannot quantify the total moment (/magnetization) only considering the LAO film layer thickness (Fig. 3d). To avoid inaccurate data evaluation, in this work, we intentionally present the absolute magnetic moment of all the measured samples rather than evaluating the magnetization as the magnetic properties are not solely from the grown film layers. Also, if magnetic impurities would be the main source for the interfacial magnetism, a strong laser fluence dependence and oxygen environment would not appear with a constant magnetic response.

This work presents the magnetic properties of the grown LAO/STO samples, measured along the in-plane direction (Fig. 1c, Figs. 2a,d, and Fig. 3a). We then show that the induced cation defect moments are preferentially aligned along the out-of-plane direction (Fig. 4e), resulting in an enlargement of the total magnetic moment of the B-site cation deficient LAO/STO system. Finally, we combine the magnetic and electronic properties of the system to deliver an emergent magnetoelectric coupling effect at the interface.

2) Fig. 1c shows a diminishment of the moment with energy. This correlation is compelling. What I do not find compelling is the assertion that the Al and Ti concentration (and hence the related vacancies) change with energy. To me the Ti concentration appears constant in Fig 1b. In the case of Al, the claim that the concentration changes is based on one data point at 1.8 J/cm². Take that one point away and the entire thesis disappears.

Reply to comment: Our work demonstrates that the Al-deficient LAO layer, grown on TiO₂-terminated STO initiates the formation of magnetic moments to the neighbouring oxygen atoms. Importantly, V_{Al} defects tend to reside near the interface, resulting in atomic compensation via Ti out-diffusion, which has been directly observed in our work by atomic resolution STEM and confirmed by DFT.

Al deficiency largely varies from 16.2 %, 15.3 %, 14.0 %, and 12.5 % over a laser fluence range of 0.6 - 1.2 J/cm². In contrast, Ti atomic ratios change from 3.60±0.02%, 3.51±0.03%, 3.40±0.05% to 3.25±0.05% within the limited thickness range (up to the maximum first five LAO unit cells). The results suggest that saturation in the Ti atomic concentration of the LAO layer with a higher laser fluence, however, overall, it indeed does show a large variation when the relative moment for each defect amount in the samples is considered as illustrated in Table R1.

Given that the Ti concentration equals the amount of Ti vacancies in the STO as a consequence of Ti out-diffusion, all the defect ratios (V_{Al}, Ti_{Al}, and V_{Ti}) are approximated from the Al deficiency (50 %-Al%) with respect to a stoichiometric ABO₃ perovskite and the Ti concentration (V_{Ti}) of the LAO films (STO). The relative moment of each defect compared to the experimentally measured total moment can be estimated by combining the ratio of their theoretical local moments (0.87 μ_B/V_{Al}, 0.9 μ_B/Ti_{Al}, and 0.48 μ_B/V_{Ti}), as determined by DFT. It should be mentioned that the relative moment are indeed overestimated if they are exclusively associated with the individual defect. It is impossible to assign these moments separately with a high inaccuracy as the total

moment is delocalised over the neighbouring ions in the system. Hence, the total magnetic moment of the B-site deficient LAO/STO system is associated with the combined defect-induced effects across the interface.

Table R1.

Sample	Al deficiency (%)	Ti ratio (%)	Total m (μemu)	m for V_{Al} (μemu)	m for Ti_{Al} (μemu)	m for V_{Ti} (μemu)
0.6 J/cm ²	16.19	3.60	47.33	21.79	17.74	7.81
0.8 J/cm ²	15.27	3.51	26.64	11.34	9.98	5.32
1.0 J/cm ²	14.00	3.44	18.67	7.34	7.39	3.94
1.2 J/cm ²	12.49	3.25	17.07	6.26	7.05	3.76

Such a defect-induced magnetic ordering across the interface is significantly inhibited by the Al richness of the LAO layer. This causes a very weak magnetic moment as seen in the conventional case of LAO/STO interface, grown by 1.8 J/cm² [Salman *et al. Phys. Rev. Lett.*, **109**, 257207 (2012)]. The LAO film growth depends on the laboratory environment, *i.e.* the particular PLD chamber, and such a high laser fluence for PLD-LAO film growth on STO that has been widely reported together with an appropriate low oxygen atmosphere to create a stoichiometric/Al-rich LAO layer for the interface conductivity [Park *et al. Nat. Commun.* 1:94 (2010), Ariando *et al. Nat. Commun.*, 2:188 (2011), Sato *et al. Appl. Phys. Lett.*, **102**, 251602 (2013), and Bi *et al. Nat. Commun.*, 5:5019 (2014)]. In addition, the effect of the B-site cation defect assembly on the interfacial magnetism can be further controlled by their interactions with oxygen vacancies in a given oxygen-reduced condition. Therefore, our findings not only show a cation defect-induced interfacial magnetism with a clear cation stoichiometry dependence but also it deepens our fundamental understanding of the magnetism in LAO/STO interfaces.

3) The authors say the induced moment on Ti is important to the overall moment. Fig. 2a shows data for a moment in a sample that doesn't have any Ti in it. Is the difference between the two curves due to the induced Ti moment, and the rest is spin-polarization on O? Why wasn't XMCD of O measured, as done by others?

Reply to comment: The O *K* XMCD spectra of a B-site cation deficient LAO(001)/LAO(001) sample which has no Ti_{Al} defects were recorded using the total electron yield method (by measuring the sample drain current). Unfortunately, the fully-oxidized B-site cation deficient 9uc-LAO layers, grown on LAO(001) substrate were too insulating that charging prevented a reliable measurement of the O *K*-edge XMCD.

Instead, to clarify oxygen spin-polarization nature in the B-site cation-deficient LAO/STO system, we performed additional XMCD measurements. Figure R2 shows the O *K*-edge XMCD spectra for an oxidized B-site cation-deficient LAO/STO sample, grown by a laser fluence of 0.6 J/cm², measured in normal X-ray incidence to the film surface. A

very clear field-swap of the XMCD signal at the O K -edge of the sample demonstrates its magnetic nature. Three different contributions to the XMCD spectrum can be distinguished. The two contributions at 529 eV and 532-533 eV lie where the O $2p$ orbitals are hybridized with the Ti $3d t_{2g}$ and e_g orbitals, respectively [Lee *et al. Nat. Mater.* **12**, 703 (2013)]. This magnetic signal cannot be due to Ti^{3+} ions produced by the V_o defects because the spectral shape of the Ti $L_{2,3}$ XMCD spectrum measured on the oxidized film is very similar to that of the pure Ti^{4+} spectrum reported by Salluzzo *et al. [Phys. Rev. Lett.* **111**, 087204 (2013)]. Moreover, the O K -edge XMCD cannot only be attributed to d_{xy} -type ferromagnetism [Lee *et al. Nat. Mater.* **12**, 703 (2013)] since our data shows a polarization of the oxygen states mixed with the Ti e_g bands. We conclude that the two XMCD contributions at 529 and 532-533 eV reflect the polarization of the oxygen bands induced by both of Ti_{Al}^{4+} and V_{Ti} vacancies. The XMCD signal centred at 535.5 eV occurs in a region dominated by O $2p$ states hybridized with La $5d$, Al $3s$ and Al $3p$ empty bands and could be associated to V_{Al} defects.

Figure R2. (a) A schematic of X-ray magnetic circular dichroism (XMCD) measurement protocol. Total electron yield (TEY) detection is obtained through excitation using circularly polarized light (ρ^+/ρ^-) with normal incidence while applying an external magnetic field along the out-of-plane $\langle 001 \rangle$ directions. (b) 20 K X-ray absorption spectra at the O K -edge of a B-site cation deficient 9uc-LAO/STO sample, grown by $E_g = 0.6$ J/cm². The energy positions of the O-hybridized bands are estimated by the theoretical electronic structure calculations of the bulk LAO and STO. (c) O K -edge XMCD spectra, measured by reversing the applied fields, $H = +6$ and -6 T.

4) Fig. 2d shows that the moment decreases with increasing energy in STO film/STO substrate. The magnitude of the moment is 3/5 of the LAO film/STO, so nearly the same. The authors suggest the polar vs. not polar character of LAO/STO induces a reconstruction that promotes O spin polarization and Ti redistribution and its induced moment. In the case of STO/STO this mechanism is lacking, and yet the magnetism is nearly as strong as the LAO/STO case. I sense an inconsistency. Further the lack change of Ti concentration with energy (see Fig. 1b) suggests to me the STO/STO sample is fairly uniform, so what is special about the STO film, when the STO substrate does not show magnetism?

Reply to comment: In the the manuscript, we clarified all of the possible defects and their contributions to the magnetism in the B-site cation deficient LAO/STO system by

comparing the LAO/STO, LAO/LAO, and STO/STO samples. We cannot make a quantitative comparison of the measured moments between the heteroepitaxial LAO/STO and homoepitaxial STO/STO and LAO/LAO samples due to different atomic/ionic kinetics during film growth and vacancy defect density. This is an interesting topic but outside of scope of this paper and further work is required. Moreover, no antisite defects form in the homoepitaxial samples due to the same charge valence of the substituted elements in the appropriate cation sites. However, a strong laser-fluence dependence of the magnetic moment in the STO(001)/STO(001) samples was observed, a similar trend is seen in the LAO/STO case. This firmly indicates that the B-site cation vacancy (V_{Ti}) contributes to the creation of the magnetism (oxygen spin-polarization) in STO. This result also qualitatively unveils one of the magnetic reservoirs in the B-site cation deficient LAO/STO system, especially on the STO side, if Ti out-diffusion is driven from the STO surface to any upper film layer.

The authors say there are no impurities, yet what is the evidence for this claim? In (1) I noted the magnetization seems to be non-sensical. If there is some error related to (1), then what concentration of Fe impurities is needed to produce the magnetism? I propose an alternate hypothesis: Magnetic impurities are deposited into the films, e.g., concomitant bombardment of the chamber, and the amount of impurities increases with energy of the PLD.

Reply to comment: First, we observe an opposite behavior as proposed by the reviewer: the magnetic moment decreases with increasing laser fluence (Fig. 1c). In any case, there is no connection between laser fluence and the concentration of impurities as it is supposed to influence ablation from target only. A presence of such high amount of magnetic impurities in the single crystalline LaAlO₃ target is not realistic. Moreover, we have not observed any traces of foreign magnetic elements in the XPS spectra. These considerations exclude possibility that magnetism in our films is due to external impurities. The magnetization saturates at about 9 μ c-thick LAO, which reveals that it is indeed the interfacial effect. If it is due to the impurities, it should just increase with the volume of the films.

5) I would say “atomic charge compensation” not “atomic compensation”.

Reply to comment: We agree with the Referee’s suggestion and revised the manuscript accordingly.

Reviewer #3 (Remarks to the Author):

The manuscript 'The emergence of magnetic ordering at complex oxide interfaces tuned by defects' by Park et al. reports on magnetic properties of LAO/STO heterostructures produced with varying LAO cation stoichiometry. The authors suggest a mechanism based on the formation of B-site cation defects, namely, titanium vacancies on the STO side, and TiAl antisite defects on the LAO side of Al-deficient LAO layers to be the main driver of magnetism in the system.

The magnetism in these systems is clearly a very interesting and impactful topic to be addressed in the field. Experimental and theoretical work as well as data evaluation is appropriate and the presented work can be reproduced based on the given information. The authors present a number of intriguing experimental data, particularly, systematic stoichiometry-dependence of the magnetic signal, and support their conclusions by DFT calculations.

While I like the idea of the paper and while I am impressed by the reported data, I am more reluctant about the details in the presented argumentation. I am missing a couple of points that should be addressed in order to arrive at a full understanding of the data. Therefore, I can recommend publication of the manuscript only after careful revision.

Reply to comment: We highly appreciate Reviewer #3 for his positive comments and suggestion and the interest on our work. Following his/her questions and comments, we provide our point-by-point reply below and have revised the manuscript accordingly.

My main concerns are the following:

1. The authors argue that the magnetism at the LAO/STO interface is p-type in nature and argue that the XMCD data obtained for the O K-edge gives evidence for this. The signal however is very small and the discussion of the data in the SI is not detailed enough. In order to be convincing, the authors should provide field-dependent data, indicating that the XMCD is robust and behaving systematic with field (or at least a field-swap). If confirmed, the discussion should be moved to the main paper – as current state of knowledge (see e.g. Refs. 16,17) is that the interface shows dxy-type magnetism. It is also not clear to me how the XMCD (Fig. S7) can 'verify the magnetization of the anticipated anti-site defects' (statement on page 5) as it is lacking the sensitivity to single defect species?

Reply to comment: Following referee's suggestion, the O K-edge XMCD measurements of the oxidized B-site cation deficient LAO/STO were repeated (with much higher statistics than the data reported in the previous version of the manuscript) at 20 K at both B = +6 and -6 T applied fields. The XAS spectra were collected in groups of eight or octets ($\rho^+ \rho^- \rho^- \rho^+ \rho^- \rho^+ \rho^+ \rho^-$, where ρ^+ and ρ^- indicate photon spin parallel or antiparallel to the applied field, respectively) in order to minimize the effect of any time dependence in the X-ray beam on the measured spectra. In total, the XMCD was obtained as the difference between 40 ρ^+ and 40 ρ^- spectra, *i.e.* 5 times more than what is commonly acquired at BOREAS of ALBA for standard ferromagnetic materials. We have added the following text and new data plots (Figs. 4a,b,c) to the revised manuscript:

“To verify the magnetic nature of the B-site cation deficient LAO/STO system, O K-edge XMCD spectra for an oxidized 9uc-LAO/STO sample, grown by a laser fluence of 0.6 J/cm² is separately measured with applied fields of $B = +6$ and -6 T. A very clear field-swap of the XMCD signal at the O K-edge, presented in Figs. 4a,b,c, demonstrates its magnetic nature. Three different contributions to the XMCD spectrum can be distinguished. The two contributions at 529 eV and 532-533 eV are seen where O 2p orbitals hybridized with Ti 3d t_{2g} and e_g orbitals, respectively, are known to occur¹⁵. Note that this magnetic signal cannot be due to Ti^{3+} ions, produced by V_o defects, because the spectral shape of the Ti $L_{2,3}$ XMCD spectrum, measured on the oxidized film (Fig. S7c), is very similar to the pure Ti^{4+} spectrum, reported by Salluzzo et al.¹⁷. Moreover, the O K-edge XMCD cannot only be attributed to d_{xy} -type ferromagnetism¹⁵ since our result also reveals polarization of the oxygen states mixed with the Ti e_g bands and other hybridized bands of the cations at higher energies. Thus, we conclude that the two XMCD contributions at 529 and 532-533 eV reflect the polarization of the oxygen bands, induced by both of Ti_{Al} and V_{Ti} defects. The XMCD signal centered at 535.5 eV occurs in a region, dominated by O 2p-states hybridized with La 5d, Al 3s and Al 3p empty bands and is associated with V_{Al} defects. The observed O spin-polarization nature remains at RT as a temperature-independent characteristic (Fig. S7d). As a result, it is confirmed that the consecutive atomic charge compensation at the interface establishes magnetic orders from aligned cation vacancy–anti-site–vacancy defects as schematically illustrated in Fig. 4d. This might further reflect the enhanced delocalization of the magnetization density outward the B-site cation defect clusters as shown by DFT calculations. In other words, the total magnetic response of the B-site cation-deficient heterostructure results from an extended defect configuration at the interface, instead of isolated defects or individual layer contributions.”

The O K-edge XMCD measurements were carried out at the maximum magnetic field available in BOREAS beamline in order to maximize the dichroic signal. A systematic study of the magnetization hysteresis loop at different magnetic fields through O K-edge XMCD technique would need very long impractical acquisition times (several days) in order to achieve sufficient reliability in the experimental data for an XMCD signal smaller than 1% of the maximum intensity of the K-edge XAS.

Finally, Ti $L_{2,3}$ XMCD is element sensitive. Our Ti $L_{2,3}$ XMCD of oxidized B-site cation deficient LAO/STO shows that Ti_{Al} antisite defect has a 4+ valence.

2. The authors should address in more detail, why there is a discrepancy between transport (AHE arises at low temperature) and SQUID data (which indicates strong magnetism up to RT). As the authors argue that the magnetism is very robust also at room temperature, why is there no anomalous Hall effect observed at room temperature?

Reply to comment: The B-site cation defect-induced magnetism in the samples is responsible for the SQUID data, while the AHE originates from the 2DEG at the interfaces. The AHE originates from magnetism at the 2DEG in the LAO/STO system, which has a strong temperature-dependent characteristic [Stornaiuolo *et al.*, *Nat. Mater.*, **15** 278 (2016) and Gan *et al. Adv. Mater.*, **31**, 1805970 (2019)]. Moreover, the cation defect-induced magnetism is much stronger than the magnetism at the 2DEG interface, leading to a magnetic proximity effect around the interface. Hence, three critical factors need to be considered in order to understand the observed strong low-temperature AHE in the B-site cation-deficient LAO/STO system: (i) a temperature-independent defect-induced magnetism; (ii) a temperature-dependent weak magnetism of the 2DEG; (iii) and the enhanced magnetoelectric coupling effect between the 2DEG and the B-site cation defect-induced magnetic moments only at low temperatures via a magnetic proximity effect. To better communicate these observations, we have added the following statements to the revised manuscript and supplementary information.

- In the revised manuscript, we have added new statements as:

“Furthermore, a hysteretic AHE feature in the B-site cation-deficient LAO/STO sample was observed near zero-field while sweeping the fields (see Fig. S10). The coercive field ($B = \sim 40 - 50$ mT) of the deconvoluted R_{xy} (AHE) in the B-site cation deficient LAO/STO is consistent with the magnetic coercive field ($H_C = \sim 40 - 45$ mT), determined by SQUID magnetometry. These directly indicate a coupling effect between the defect-induced magnetic moments and the 2DEG. In addition, the hysteretic AHE of the 2DEG visibly appears up to 5 K. This is in contrast to a strong temperature-dependent hysteretic AHE feature in conventional 2DEG interfaces, usually limited by very low temperatures ($T < 2$ K) in previous reports⁴⁰⁻⁴³. These observations further confirm that the weak magnetism of the 2DEG reported until now can be effectively coupled and enhanced by much stronger B-site cation defect-induced magnetism. Therefore, we assign the observed large enhancement in the AHE to a magnetic proximity effect between the conduction electrons of the 2DEG and the B-site cation defect-induced magnetic moments normal to the interface.”

- In the supplementary Information, we have added new statements:

“To clarify the coupling effects between the defect-induced magnetic moments and the 2DEG near zero field, magnetotransport measurements of a B-site cation-deficient LAO/STO sample were carried out with refine field intervals (1 – 2 mT/sec) considering the measurement time (/field interval) dependence of the ferromagnetic coercive strength of the system. As seen in Figs. S10b,e, a clear hysteretic R_{xy} appears at the zero field while sweeping the fields. This corresponds to an increase in

the Hall coefficient, R_H , towards zero field and a spiky negative R_H then occurs when the applied field is reversely switched across the zero field due to the magnetic remanence of the system (Fig. S10f). The coercive field ($B = \sim 40 - 50$ mT) of the deconvoluted R_{xy} (AHE) part of the B-site cation deficient LAO/STO is consistent with the magnetic coercive field ($H_C = \sim 40 - 45$ mT), determined by SQUID magnetometry (Fig. S10g). In addition, the hysteretic AHE of the 2DEG visibly appears up to 5 K. These observations further confirm that the weak magnetism of the 2DEG reported until now can be effectively coupled and enhanced by much stronger defect-induced magnetism.’’

3. The authors do not address the dynamics of the proposed defect formation process – the diffusion of B-site cations in perovskites is typically very slow hindering the formation of B-site vacancy defects (and often favoring A-site defect formation). Also, the formation energy for B-site vacancies is typically large (e.g. Akthar, Catlow, J. Am. Ceram. Soc. 78, 421 (1995))? It would be important the authors address in more detail how is the Ti moving into the LAO layer to form anti-site defects? Also the charge-state of the TiAl anti-site defect should be discussed – where is the extra electron in the LAO layer going?

Reply to comment: To address this, we need to understand the difference between thermodynamic equilibrium growth condition and the non-equilibrium conditions for solid-state oxide thin film formation. Any artificially designed epitaxial complex oxide films are usually made in non-equilibrium states and/or in quasi-states. This is one of the most significant advantages of using PLD for designing/controlling atomically defined thin film layers on underlying substrates by adjusting material’s stoichiometry to trigger exotic physical phenomena, compared to conventional properties in bulk materials. Together with this growth feature, the stoichiometry and related properties of the designed LAO/STO thin films can differ from those of bulk materials.

Akthar *et al.* [*J. Am. Ceram. Soc.* **78**, 421 (1995)] computed the defect formation energies and diffusion activation energies in bulk STO at the equilibrium conditions and $T = 0$ K. As the result, the computed Ti activation energy of ~ 11.6 eV is significantly overestimated and, hence, it cannot be directly discussed with the context of our results, especially because Ti diffusion goes into the LAO overlayers across the interface.

Meanwhile, we have found that Ti out-diffusion is significantly limited to a maximum of 5uc of the LAO upper layer in the most B-site cation deficient sample in this work. The Ti diffusion can be effectively hindered by the reduction of the Al vacancy concentration of the LAO layer. This is what we have directly observed in our experimental results as a function of laser fluence. An overall dynamic picture of the consecutive defect assembly is presented in Fig. 4d in the main text of our manuscript.

4. I think it is very interesting finding that homoepitaxial STO and LAO can both show magnetism when grown in a non-stoichiometric manner, which nicely scales with layer thickness. It is not clear for me (this may be a matter of clarity only), why for LAO/STO the magnetization saturated, as I would always expect to see the thickness-dependent

magnetization of the thin film on top of the one at the interface? The authors argue that in the heterostructure parts of the cationic defects are co-compensated by oxygen vacancies. This is a straight forward assumption, but why is it not happening in the homoepitaxial thin films, which following this arguments should not show magnetization?

Reply to comment: We thank the reviewer for appreciating the results. The total magnetic moment of the B-site deficient LAO/STO system stems from the combined defect assembly effect across the interface. As we proposed in the manuscript, by compensating defect formation, the defect level of B-site cation vacancies in LAO can be saturated according to the amphoteric defect theory. Furthermore, an important point made by the referee is that cation diffusion is usually slow. So, Ti out-diffusion to the upper film layer is limited by the LAO layers (e.g. the maximum first 5 ucs in the B-site cation deficient LAO/STO sample, grown by $\sim 0.6 \text{ J/cm}^2$ laser fluence), as also observed by STEM.

We agree with the referee's comment, and we also expect a similar saturation behaviour in the magnetic moment of homoepitaxial LAO or STO films which would appear with thickness. However, the magnetism is mainly due to B-site cation vacancies as the same valence charge of substitutional cations are always observed around the interfaces.

5. The authors rule out other defect scenarios (oxygen vacancies and Sr vacancies) based on DFT, but they do not show their data (page 4, bottom). This should be done at least in the SI as both statements are surprising – oxygen vacancies have been identified in the literature to be a possible origin of magnetism in STO (see e.g. Refs. 17/18). For Sr vacancies one may expect a similar p-type hole formation as observed in the Ti vacancy case – it is hence interesting to understand in how far they result in different behavior?

Reply to comment: We did try to search for any oxygen vacancy effect in the magnetism of STO, but this was not the case.

Firstly, DFT shows that V_o -induced magnetism in STO strongly depends on the concentration of V_o for the magnetization of Ti d_{xy} orbitals as shown in Figs. R3a,b. Our calculations show that in a (4×4) STO supercell, the magnetic moment of a single V_o (~ 2 at.%), separated by the distance of $4a$ (where a is the lattice constant) becomes a negligible value of $0.003 \mu_B/V_o$. In contrast, a higher V_o concentration (~ 8 at.%), formed as a regular sublattice in the STO by reducing the distance to $2a$, effectively magnetizes the Ti d_{xy} bands, similar to reported results. This results in $0.4125 \mu_B/V_o$ after relaxation. Recently, a similar feature on the magnetic double exchange interactions between magnetic species Ti^{3+} and Ti^{4+} has been demonstrated [*Sci. Rep.* **5**, 7909 (2015)]. However, the V_o concentration in a modelled structure by authors of this paper seems unrealistically high. Also, practical existence of oxygen vacancies in our sample could not be the case as all of the magnetic B-site cation deficient LAO/STO samples were oxidized after film growth (except non-oxidised conductive samples which were used for magnetotransport measurements).

Secondly, the formation of La vacancy defects in the LAO film layer of magnetic B-site cation deficient LAO/STO system can be disregarded, based on our experimental results (e.g. Al/La atomic ratios).

Finally, an STO supercell model structure with a Sr vacancy was calculated by DFT. Unlike the B-site cation vacancy effect, there is indeed a very weak A-site cation vacancy effect ($m < 0.07 \mu_B/V_{Sr}$) on the creation of magnetism in STO (Fig. 3c) although the defective system shows a *p*-type characteristic. This could be due to much less charge rearrangement of the neighbouring oxygens and cations near the Sr vacancy due to the following factors: (i) only two electrons can contribute to neighbouring oxygens at the Sr vacancy sites, compared with four for the Ti (4+) cation vacancy; (ii) the Sr site has 12 nearest oxygens at relatively larger distances of 2.76 Å, whereas Ti has 6 oxygen neighbours at significantly shorter distances of 1.95 Å. Therefore, the Sr vacancy (or A-site cation vacancies in ABO_3 perovskites) would not dramatically change the electronic structure. This eventually results in a magnetic moment at each oxygen atom of less than 0.001 μ_B .

Figure R3. (a) Magnetization density of a (4×4) STO supercell with an oxygen vacancy (~ 2 at.%), separated by a $4a$ distance. a is the lattice constant of STO. The spin charge imbalance is presented by yellow (positive) and cyan (negative) colour. (b) Magnetization density of a (2×2) STO supercell with oxygen vacancies (~ 8 at.%), separated by $2a$ distance. (c) The total spin-polarized density-of-states (DOS) for an STO supercell with a Sr vacancy (in the upper panel) and the LDOS of the nearest neighbouring O (in the middle panel) and Ti (in the lower panel) atoms next to the Sr vacancy.

6. A quantification for the magnetic data should be given. Based on SQUID, one should estimate the number of anti-site defects at the interface.

Reply to comment: We thank the reviewer for his/her valuable comment, however a quantification analysis is very challenging for assigning specific magnetic moments to each defect in the B-site cation deficient system as the total sample magnetization originates from a combined magnetic effect. Also, the entire defective layer across the interface (not only the film layer) needs to be considered. For example, single ion pictures are often described by assigning a moment per Mn ion in the transition metal-

based magnetic material system (e.g. $\text{La}_{1-x}\text{Sr}_x\text{MnO}_3$ films), but it is not accurate as magnetization is often delocalised over different ions.

We cannot determine practically specifically how much effective moment is from each B-site cation defect with a gradient defect distribution and a moment spread across the interface (as seen in STEM) although their moments have been determined by DFT. Also, the total moment is distributed over all of the neighbouring ions. A magnetic moment of around $1.8 \mu_{\text{B}}/\text{uc}$ for the most B-site cation defective LAO/STO system (grown by $\sim 0.6 \text{ J}/\text{cm}^2$ laser fluence) could be deduced by considering the depth of defect distribution (about 16uc thickness) and experimental total magnetic moments. When the magnetic moment of 9uc-LAO upper layers is considered by combining the Ti atomic ratio and estimated defect ratios of V_{Al} , Ti_{Al} , and V_{Ti} in the B-site of the perovskite cell, it could be further assigned to be 2.6, 1.6, 1.1, 0.97, and $0.14 \mu_{\text{B}}/\text{uc}$ with the laser fluence of $0.6 - 1.8 \text{ J}/\text{cm}^2$. Obviously, this moment density variation indicates it is not simply a magnetic impurity effect (e.g. $3.6 \mu_{\text{B}}/\text{Fe}$). Moreover, an estimated concentration of the out-diffused Ti atoms is $\sim 6.2 \times 10^{20} /\text{cm}^3$ in the most B-site defective LAO layer. The corresponding Ti_{Al} defect magnetic moment from the portion of SQUID moment ($210 \text{ emu}/\text{cm}^3$) is deduced to be $36.4 \mu_{\text{B}}/\text{Ti}_{\text{Al}}$. The deduced moment per single defect is unrealistic. Thus, it is noted that each B-site cation defect cannot solely contribute to each portion of the measured moment with respect to B-site cation defect ratios in LAO layer. This means it is difficult to assign moments to single defects, which is not reliable with quantitative inaccuracy. In other words, the obtained large total magnetic moments should be interpreted as collective/combined magnetic moments as a result of a collective magnetic interactions in the B-site cation defect assembly. One of our ongoing projects is to address these issues. Therefore, at this stage, our work only demonstrates a qualitative picture to describe the effect of B-site cation defect assembly across the interface combining experimental results, supported by theoretical model calculations.

Besides these major comments, I have some minor comments the authors should consider:

- On page 3, I would recommend to mention the growth pressure in the main text, as it is important for the required plasma-gas interaction responsible for the stoichiometry variation.

Reply to comment: We agree with the reviewer and according to the referee's suggestion, we have added statements in the revised manuscript as:

“The laser fluency (energy per unit area) and plume-background gas interaction significantly affect the transfer of ablated species to the surface of underlying substrates during deposition, resulting from the non-congruent ablation and/or mass-dependent variations in the atom and ion trajectories through the laser plume.”

- Page 3: The term ‘angle-dependent’ XPS is misleading as measurements were made only at one angle. I suggest to remove. Also, the information depth should be mentioned for the individual core levels. As the authors argue in the SI, the information depth will be different for each element/core level.

Reply to comment: We agree with this, so the term, “angle-dependent XPS” has been

corrected to be “*surface-sensitive XPS*” in the revised manuscript. In addition, to inform the inelastic mean-free path of photoelectrons for constituent element core levels of LAO and STO, we have added the details in the revised manuscript and the Method section as:

“*Figure 1b shows the atomic composition of the as grown LAO/STO samples, determined by a surface-sensitive (a take-off-angle (TOA) of 30 degrees) X-ray photoelectron spectroscopy with effective probing depths of approximately max. 3.2 – 5.6 nm for all the constituent element core-levels of LAO and STO. (see the Method section and S2).*”

“*The maximum probing depths of the surface-sensitive XPS for Al 2p, La 4d, Sr 3p, and Ti 2p core levels of LAO and STO, collected with a 30 degrees TOA, were estimated to be ~4.7, ~5.6, ~5.6, and ~3.3 nm, respectively.*”

• Page 3: ‘reduction of Al’ – I would suggest to use ‘reduction of concentration of Al’ to avoid confusion with a valence-change process.

Reply to comment: We have corrected this in the revised manuscript according to the suggestion as:

“*i.e. the reduction of the concentration of Al while increasing of La (Fig. 1c).*”

• Fig. 1c – can the authors comment on why the substrate gives a paramagnetic signal, while for the LAO/STO samples a diamagnetic background is observed?

Reply to comment: Diamagnetic and paramagnetic background components in the measured magnetic data are from an instrumental accessory (sample holder) in the SQUID magnetometer, used in this work. Therefore, we have corrected the data again as Fig. 1c in the revised manuscript.

• The section on transport analysis and the data treatment is mainly following Ref. 40, which should be indicated more clearly in the main text. Also, a reference to Joshua, Nat. Comms. 3, 1129 (2012) should be added.

Reply to comment: We agree with Referee’s suggestion, so the suggested paper is added to the list of references in the revised manuscript:

“39. Joshua, A., Pecker, S., Ruhman, J., Altman, E. & Ilani, S. A universal critical density underlying the physics of electrons at the LaAlO₃/SrTiO₃ interface. Nat. Commun. 3, 1129 (2012).”

• Fig. 3h – data seems to be not consistent to Fig. 2a/3a? Why is the saturation magnetization in perpendicular field much bigger than anticipated in 2a/3a?

Reply to comment: The samples presented in Figs. 2a/3a were grown by different laser fluence as stated in the manuscript. The LAO thin films in Fig.2 were grown with a laser energy fluence of ~0.6 J/cm², while for the thickness-dependent series, the films were grown using a laser energy fluence of ~0.7 J/cm². As for the second question, the magnetic moments of the ordered cation defects along the out-of-plane of the B-site

cation-deficient LAO/STO sample could be more collective as a result of the consecutive defect formation across the interface.

• Fig. 4 – it is very interesting that the non-linearity of normal Hall effect and AHE almost cancel out, resulting in an almost linear R_{xy} . Can the authors comment how stable the fitting of the data is in this case?

Reply to comment: We agree with the referee that the nonlinear parts of the normal Hall effect (assuming two-carrier case) could be cancelled by anomalous Hall effect (AHE). Based on our experimental and fitting results, this feature visibly appears as illustrated in Figs. R5a,c when R_{xy} is subjected to a linear ordinary Hall effect. The key point relies on the critical field (B_{FWHM}) where the nonlinear parts are significantly different in nature as shown in Fig. R4. Here, the non-linearity of the normal Hall effect for the two-carrier case occurs at high magnetic field (after a critical field), leading to a Lorentz-shaped R_H curve. However, the non-linearity of AHE occurs at low magnetic field, leading to a hump around zero field if a non-hysteretic feature appears (Fig. R4).

Following the referee’s comment, we present additional statements and new plots for the NOHE and AHE parts of the experimental Hall resistance in the Supporting Information S9:

“Here, the non-linearity of the normal Hall effect stems from two-carrier case, leading to a Lorentz-shaped R_H curve. The corresponding Hall coefficient can be expressed as:

$$R_H(B) = \frac{R_{xy}(B)}{B} = R_{xy}(\infty) + \frac{R_H(0) - R_H(\infty)}{1 + (2B/B_{FWHM})^2},$$

where $R_H(0)$ and $R_H(\infty)$ are the values of $R_H(B)$ at $B \rightarrow 0$ and $B \rightarrow \infty$ limits, respectively. The B_{FWHM} also describes the full width at half maximum of the Lorentz-shaped R_H curves.

$$B_{FWHM} = 2 \frac{n_1 \mu_1 + n_2 \mu_2}{(n_1 + n_2) \mu_1 \mu_2},$$

Figure R4. NOHE and non-hysteretic AHE fitting parts of the experimental Hall coefficient, $R_H (= R_{xy}/B)$, of a B-site cation deficient LAO/STO, measured at 2 K. The AHE of the sample can strongly perturb the NOHE over the applied field: the AHE part of R_H increases toward the zero field, opposite to the Lorentz-shaped NOHE part.

Figure R5. (a) Linear ordinary Hall effect (LOHE) and anomalous Hall effect (AHE) fitting parts of the experimental 5 K-Hall resistance [R_{xy} (Exp.)] of a B-site cation deficient LAO/STO sample, grown with $E_g = \sim 0.6$ J/cm², measured in a magnetic field of 5 T with a field sweep of 1 - 2 mT/sec. (b) The remanent R_{xy} (Exp.) of the sample at the zero field while sweeping the fields. (c) Residual S-shaped R_{xy} over the applied field and hysteretic AHE in small field range in the deconvoluted AHE part after subtracting the LOHE part ($R_{xy} = R_0B$) to the total R_{xy} (Exp.). (d) Nonlinear ordinary Hall effect (NOHE) and AHE fitting parts of the R_{xy} (Exp.) of the sample. (e) The deconvoluted hysteretic AHE part and the difference of the forward and reverse R_{xy} (Exp.) while sweeping the fields at 5 K. $R_{xy}(F)$ was measured in a forward field sweep (+5 T to -5 T) and $R_{xy}(R)$ was measured in a reverse field sweep (-5 T to +5 T). (f) The hysteretic AHE part of the R_H (Exp.) of the sample, measured in the field range of 5 T. The hysteretic AHE part visibly appears in the applied field below 1 T. Correspondingly, a spiky negative R_H occurs while the applied fields cross over the zero field and the negative R_H remains up to $B \sim 40$ - 50 mT. (g) A 5 K out-of-plane magnetic hysteresis loop (in the upper panel) and the deconvoluted AHE part (in the lower panel) of the R_{xy} of the sample. Dashed lines indicate the coercive field ($H_C = \sim 45$ mT) of the magnetic moment and R_{xy} while sweeping the applied fields.

• Page 6: It is not clear to me, why a proximity effect of the hysteretic out-of-plane magnetization in the LAO layer on the 2DEG does not yield a hysteresis in the observed AHE?

Reply to comment: To address this, we have performed additional low temperature magnetotransport measurements of a B-site cation deficient LAO/STO sample ($E_g = 0.6$ J/cm²) with small field intervals (5 -20 mT) which are much smaller than the magnetic coercive field ($H_C = 40 - 45$ mT). Our new results visibly show a hysteretic AHE of the measured R_{xy} near the zero field while sweeping the field in a closed loop. Therefore, we have added the new results in the revised supplementary information (Figs. S10):

“To clarify the coupling effects between the defect-induced magnetic moments and the 2DEG near zero field, magnetotransport measurements of a B-site cation-deficient LAO/STO sample were carried out with refine field intervals (1 – 2 mT/sec) considering the measurement time (/field interval) dependence of the ferromagnetic coercive strength of the system. As seen in Figs. S10b,e, a clear hysteretic R_{xy} appears at the zero field while sweeping the fields. This corresponds to an increase in the Hall coefficient, R_H , towards zero field and a spiky negative R_H then occurs when the applied field is reversely switched across the zero field due to the magnetic remanence of the system (Fig. S10f). The coercive field ($B = \sim 40 - 50$ mT) of the deconvoluted R_{xy} (AHE) part of the B-site cation deficient LAO/STO is consistent with the magnetic coercive field ($H_C = \sim 40 - 45$ mT), determined by SQUID magnetometry (Fig. S10g). In addition, the hysteretic AHE of the 2DEG visibly appears up to 5 K. These observations further confirm that the weak magnetism of the 2DEG reported until now can be effectively coupled and enhanced by much stronger defect-induced magnetism.”

• XPS survey scans in SI – there are two core levels labeled with La 3d.

Reply to comment: In the revised manuscript, “La 3d” is corrected to be “La 4d” at the binding energy of approximately 99 – 110 eV.

• XAS O K edge: the authors should indicate also positions of Sr hybrids for clarity.

Reply to comment: In the O K-edge XAS of the 9uc-LAO/STO, the spectral features of Sr-O hybrid orbitals, coming from the STO substrate buried under LAO, could be difficult to observe (with a very small spectral weight) and only features from La- and Al-O hybridized bands (plus Ti 3d –O 2p bands) dominate the spectra. To shed light on the energy positions of Sr-O hybrids, Figure R6 shows the XAS of the B-site cation deficient LAO/STO together with the XAS of STO. The energy positions of each O-hybridised band are estimated by LDA calculations.

Figure R6. The XAS spectra of the O K -edge B-site cation deficient LAO/STO (lower) and STO (upper). The energy positions of the O-hybridized bands are estimated by the LDA model calculations of the bulk LAO and STO.

Reviewers' comments:

Reviewer #1 (Remarks to the Author):

I am still lost by the arguments made by the authors concerning the “second” part of the manuscript that claims to make a non-trivial connection between the magnetism due to the B-site cation defects (which are firmly established) and any magnetic properties in the STO. I feel the need to reply in-line to the brief response of the authors:

“I feel that there are really two distinct sets of results reported in this paper. I would support in principle publication of a revised manuscript that sticks to the cation vacancy data and analysis. The second part of the manuscript I do not feel rises to the same level of significance or clarity.

Reply to comment: We thank the reviewer for his/her comments and we therefore tried to make our arguments clearer. Our findings show that interfacial magnetism in LAO/STO system can be effectively tuned by the distribution of B-site cation defects.

On this we are in agreement.

We have further showed that the defect-induced interfacial magnetism can be significantly coupled with the interface conductivity, primarily created by the presence of oxygen vacancies. Hence, our results provide clear evidence and methodology of designing defect assembly and tuning the interfacial magnetism.”

Here I am still not convinced. I was pleased to see the new measurements confirming existence of hysteresis in the AHE. It is not clear to me, however, that it is anything other than the contribution of the magnetic field from external magnetic moments that were previously identified. It is not clear what the authors are referring to as “weak magnetism”. This measurement basically confirms that there is no intrinsic hysteretic (ferromagnetic or ferrimagnetic) behavior in the STO. The hysteretic behavior is no more surprising than if any other ferromagnetic layer were placed in contact with the top surface and switched hysteretically.

The existence of oxygen vacancies and their impact on transport in the STO layer are a completely separate matter. There is absolutely nothing novel here concerning the use of oxygen vacancies to tune the electron density in LAO/STO. The final sentence of the author’s response is also an overreach. There is no evidence that the effect of the magnetic moments on the STO is any different from the externally applied magnetic field.

It was my recommendation previously that the authors scale back their discussion to focus on the magnetic moments that they were able to create and control using oxygen pressure and laser fluence. I would have recommended publication of that manuscript. Here, these interesting results are mixed with overreaching claims that are not supported by the data. Unfortunately, I cannot recommend the manuscript for publication in its present form.

It is not clear that the hysteresis is gone even in the 1.8 J/cm² sample. The vertical scale should be expanded to reveal whether there is any hysteresis. To my eye, it looks like the sample is still hysteretic, but with a reduced density of magnetic moments.

Reviewer #2 (Remarks to the Author):

I recognize the authors have been responsive to most comments from me and the other reviewers. The O XMCD is a valuable addition.

My first takeaway from this paper is that PLD grown films can be very defective, even homoepitaxial films.

Reporting only the moment, sidesteps the issue that the moment density is likely 25% of an iron film (for STO on STO) and to me this raises concerns.

The conclusion says the B-site cation defects are the main source for magnetism in LAO/STO and STO/STO. My second takeaway is that the polarization catastrophe is irrelevant with respect to magnetism in LAO/STO and STO/STO. I'm comfortable with this interpretation of the work. Is this interpretation consistent with the authors' intent?

The authors should be made to show error bars on the data in Figure 1b. The authors claim in the rebuttal the trend is significant, without error bars I think not.

I recommend publication.

Reviewer #3 (Remarks to the Author):

The authors have made large efforts to improve the manuscript. Although some arguments given by the authors in the response letter are not yet fully clear in my eyes, I see the manuscript significantly improved. From the mechanistic view, however, the main novelty of the paper are the general mechanisms that may generate magnetism in STO and LAO. The mechanism at the LAO/STO interface, as the authors state themselves, is still hard to be distinguished from other contributions. Also my co-referees had a similar impression.

Personally, I think the observations of a) the stoichiometry dependence of magnetism in LAO and STO and b) the oxygen dichroism in STO (including the clear field swap presented in the revised manuscript) are most striking and deserve publication. As for the LAO/STO interface, the most striking result is the observation of the hysteretic anomalous Hall effect, revealed in the revised version of the manuscript. At the same time this hysteresis raises some new questions as listed below. I can therefore only recommend publication after another revision.

- The hysteretic AHE is highly interesting and could be worth to be moved into the main paper. The authors should comment, however, why the magnetic moment causing the proximity effect seems to be out-of-plane? This is counter-intuitive, as one may expect any moment to align in-plane given the thin layer thicknesses and the 2D-character of the interface.

- It may be technical, but it is unclear from the experimental point of view how the data for forward and backward field sweep was obtained. As described the samples were measured in van der Pauw configuration. Were the data averaged over different geometries to avoid misalignment voltages? And how does the potential data treatment affect the hysteretic Hall signal, which is symmetric in field and could be averaged out through averaging. Further, it is unclear how the authors technically determined the coercive field from the data.

- Finally, the model applied to describe the AHE in the RH data (elaborated in the SI) is symmetric in B, while RH now has asymmetric contributions due to the observed hysteresis. Can the authors comment why the model is still valid to be used?

- O-K-edge XMCD: While I appreciate the clear field swap shown in the new figure 4, I would like to ask why the feature 535eV cannot be related to oxygen-Sr hybrids? In my understanding, the total electron yield in XAS is not similarly surface sensitive as e.g. XPS, so they spectra may contain significant contributions from the substrate in this region. Can the authors clarify?

- I also think the statement on the Ti XMCD originating from Ti_{Al} defect sites should be rephrased. Obviously, XMCD is element sensitive. Hence, the authors observe XMCD originating from Ti with no doubt. Their conclusion the respective Ti ions are located in the LAO layer forming anti-sites however cannot be made based on the XMCD measurement alone. The authors should clarify this to avoid misunderstanding.

- As for the defect dynamics, I see the argument made by the authors, but I would suggest to comment on this in the paper. It is not obvious that anti-site defect can be formed under the non-equilibrium conditions of PLD, and should hence be mentioned in the manuscript.

- Finally, a quantitative estimation of a magnetic moment per defect – be it anti-site, cation vacancy, or something else should be made. I understand that exact values for each potentially involved magnetic center cannot be determined, but it should be made plausible that the measured total moment is consistent with the discussed (superimposed) mechanisms. A similar plausibility argument was also requested by reviewer #1.

Reviewer #1 (Remarks to the Author):

We thank the reviewer 1 for his/her additional comments and concerns to improve our manuscript.

I am still lost by the arguments made by the authors concerning the “second” part of the manuscript that claims to make a non-trivial connection between the magnetism due to the B-site cation defects (which are firmly established) and any magnetic properties in the STO. I feel the need to reply in-line to the brief response of the authors:

“I feel that there are really two distinct sets of results reported in this paper. I would support in principle publication of a revised manuscript that sticks to the cation vacancy data and analysis. The second part of the manuscript I do not feel rises to the same level of significance or clarity.

Reply to comment: We thank the reviewer for his/her comments and we therefore tried to make our arguments clearer. Our findings show that interfacial magnetism in LAO/STO system can be effectively tuned by the distribution of B-site cation defects.

On this we are in agreement.

We have further showed that the defect-induced interfacial magnetism can be significantly coupled with the interface conductivity, primarily created by the presence of oxygen vacancies. Hence, our results provide clear evidence and methodology of designing defect assembly and tuning the interfacial magnetism.”

*Here I am still not convinced. I was pleased to see the new measurements confirming existence of hysteresis in the AHE. It is not clear to me, however, that it is anything other than the contribution of the magnetic field from external magnetic moments that were previously identified. **It is not clear what the authors are referring to as “weak magnetism”.** This measurement basically confirms that there is no intrinsic hysteretic (ferromagnetic or ferrimagnetic) behavior in the STO. The hysteretic behavior is no more surprising than if any other ferromagnetic layer were placed in contact with the top surface and switched hysteretically. The existence of oxygen vacancies and their impact on transport in the STO layer are a completely separate matter. There is absolutely nothing novel here concerning the use of oxygen vacancies to tune the electron density in LAO/STO. The final sentence of the author’s response is also an overreach. There is no evidence that the effect of the magnetic moments on the STO is any different from the externally applied magnetic field.*

Reply to the reviewer comments: Since the reviewer is still puzzled by the “second” part we will try making our argument clearer by introducing also the “first” part of the discussion. The overall magnetism in our LAO/STO samples are large with the B-site cation deficient LAO overlayer. The magnetic properties are created by tuning the atomic charge compensation process which occur across the interface. In the 1st case (samples fabricated and annealed after the deposition in high oxygen partial pressure in the chamber) the magnetic moments come from the contribution of the defects at LAO layer V_{Al} and Ti_{Al} as well as defects created at the STO interfaces, V_{Ti} . This

magnetic contribution of the B-site cation vacancy has been qualitatively verified and presented in Figs, 2a-c. Note, as expected, no evidence for oxygen vacancies were observed (samples are nonconductive).

In the 2nd case (samples fabricated and cooled after the deposition in lower oxygen partial pressure) we believe again that the magnetic contribution (lower than the 1st case) comes from the same sources, defects at LAO layer V_{Al} and Ti_{Al} as well as defect created at the STO interfaces, V_{Ti} . However, here we indeed observed oxygen vacancies (samples show conductivity) however, their contribution to the magnetism is in a “negative” way as they compensate with the p -type defects, thus reducing the magnetism. This magnetic contribution of the B-site cation vacancies has been qualitatively verified and presented in Figs, 2d-f.

The first set of results agree well with the observation reported by Ariando’s group [Ariando, *et al. Nat. Commun.* **2**, 188 (2011)] where enhanced magnetism is observed in the case of samples treated in high oxygen partial pressure as compared to lower oxygen partial pressure where the magnetism is reduced. The B-site cation defect-induced magnetic moments dominantly gives rise to polarizing the 2DEG via a magnetic proximity effect, resulting in the observed large AHE. Furthermore, we emphasize here that the observed large magnetoelectric coupling effect can be driven by the large magnetic moment of B-site cation defects in pure LAO/STO system without employing external magnetic elements/layers.

Regarding the weak magnetism at the 2DEG-interface, we recognized that there is miscommunication in the manuscript, which we have now revised.

“At low temperatures (≤ 50 K), the motion of the confined d_{xy} electrons couples with the preferential out-of-plane magnetic moments”.

The out-of-plane magnetic moments are induced by the B-site cation defects, it does not mean that 2DEG itself has out-of-plane magnetism. This have been corrected in the manuscript:

“At low temperatures (≤ 50 K), the motion of the confined d_{xy} electrons couples with the preferential out-of-plane magnetic moments of the B-site cation defects, resulting from a magnetic proximity effect.”

It was my recommendation previously that the authors scale back their discussion to focus on the magnetic moments that they were able to create and control using oxygen pressure and laser fluence. I would have recommended publication of that manuscript. Here, these interesting results are mixed with overreaching claims that are not supported by the data. Unfortunately, I cannot recommend the manuscript for publication in its present form.

Reply to the comments: In principle, we could agree with the reviewer’s idea of separating our manuscript into two parts. However, we also need to balance the

concerns of other the reviewers who are highly interested in the corresponding large magnetoelectric coupling effect around the interface, induced by the B-site cation defect-associated magnetism. In fact, the other two reviewers do not recommend separating the paper into two part. The magneto-transport data of the 2DEG-interface provide an additional experimental proof for the existence of the B-site cation defects with magnetic nature. We do believe that these findings will be interesting for readers with a broader scope of research, such as the other two reviewers.

To try to comply with the reviewer request/wish, we have shortened the second part in the revised manuscript and remove some of the discussion and results to the supplementary information for better illustrating the completion of the manuscript.

It is not clear that the hysteresis is gone even in the 1.8 J/cm² sample. The vertical scale should be expanded to reveal whether there is any hysteresis.

Reply to the comments: As requested by the reviewer, we have also rescaled the magnetic moment of 1.8 J/cm²-sample in Fig. 1c of the revised manuscript in accordance with these concerns.

Reviewer #2 (Remarks to the Author):

We highly appreciate the reviewer's positive feedback and his/her recommendation for publication.

My second takeaway is that the polarization catastrophe is irrelevant with respect to magnetism in LAO/STO and STO/STO. I'm comfortable with this interpretation of the work. Is this interpretation consistent with the authors' intent?

Reply to comments: Yes, our findings show that the magnetism of the LAO/STO system as well as the STO/STO system can be tuned effectively by controlling the B-site cation stoichiometry. Indeed, the cation defect-induced magnetism is independent of the effect of polarization discontinuity at LAO/STO interface.

The authors should be made to show error bars on the data in Figure 1b. The authors claim in the rebuttal the trend is significant, without error bars I think not.

Reply to comments: We have complied with the request of the reviewer and included in Figure 1b an error bar for each data point. To further visualize this, we have revised the graph again, now including the error bars.

Reviewer #3 (Remarks to the Author):

We thank the reviewer 3 for his/her additional comments for improving our manuscript.

Personally, I think the observations of a) the stoichiometry dependence of magnetism in LAO and STO and b) the oxygen dichroism in STO (including the clear field swap presented in the revised manuscript) are most striking and deserve publication. As for the LAO/STO interface, the most striking result is the observation of the hysteretic anomalous Hall effect, revealed in the revised version of the manuscript. At the same time this hysteresis raises some new questions as listed below. I can therefore only recommend publication after another revision.

Reply to comments: We thank the reviewer for acknowledging the results of the paper and for recommending it for publication after this revision.

*- The hysteretic AHE is highly interesting and could be worth to be moved into the main paper. The authors should comment, **however, why the magnetic moment causing the proximity effect seems to be out-of-plane?** This is counter-intuitive, as one may expect any moment to align in-plane given the thin layer thicknesses and the 2D-character of the interface.*

Reply to comments: We thanks the reviewer for the good suggestion and we have moved the hysteretic AHE part into the main text. Also, we have revised the second part of the manuscript and supplementary information to deliver a clearer message for readers.

The main question raised by the reviewer is why the cation defects create preferential out-of-plane magnetic moments as experimentally observed in SQUID, XMCD, and magneto-transport measurements. The magnetic anisotropy of LAO/STO system is often describe using a magnetic model including the oxygen vacancies in TiO₂ interfacial layer. For instance, magnetic Ti³⁺ ions appear on a regular sublattice after removing each second oxygen from that layer (~50% oxygen vacancy concentration), which can interact ferromagnetically along the in-plane, thus their magnetic moments could align to the in-plane. In thin films with cubic symmetry, the shape anisotropy effect usually forces magnetic moments to be in the in-plane. In our case, the situation is different, the magnetic LAO thin layer is tetragonally compressed in the B-site cation deficient in the LAO/STO system, which can induce the uniaxial magnetic anisotropy as seen in our measurements. Furthermore, the cation defect-induced magnetic configurations in the B-site cation deficient LAO/STO play their own role and forms the preferential out-of-plane magnetic axis below:

In the manuscript, we have suggested a possible mechanism for the preferential out-of-plane magnetic moment of the B-site cation defects in the system with an exchange interaction of electrons around the consecutive cation defects, dipole-dipole interactions, and a planar orbital magnetism. In brief, these defects are characterised by unpaired *p*-electrons of O, which surround an Al or Ti vacancy, or by partially occupied Ti 3*d* electrons in the LAO environment. These electrons are

“almost” free to move within a confined space (cavity) around the defects. These cavities are magnetic due to the unpaired O p -orbitals or Ti $3d$ -electrons and interact magnetically with each other. Using the magnetic force theorem we estimated the exchange interaction between the cavities. In the LAO/STO interfaces this interaction leads to a ferromagnetic order. The magnetic interaction within a cavity results in a strong effective magnetic field (Weiss field), which forces almost free electrons to move inside the cavity - it is an orbital motion perpendicular to the field direction. This motion leads to a large orbital magnetic moment, which depends on the cavity size and shape, the number of free electrons within the cavity and their spin magnetic moments, the exchange interaction between the moments, and the temperature. For example, the total magnetic moment depends strongly on the cavity size, which could reflect the magnetic moment variation with the B-site cation defect clustering in the system. We believe that this model explains two important facts of our experiments: (i) an enhanced magnetisation due to the orbital motion and the (ii) dependency of the magnetisation on the applied field direction. We also believe that the cavity shape is not spherical along the interface - the asymmetry is due to strong relaxations in the interface vicinity: the cavities are extended along the interface and are squeezed in the direction perpendicular to the interface. Therefore, the total magnetic moments are larger for the case if the applied magnetic field is parallel to Z -axis. However, the theoretical description and details are beyond the scope of this work and can be a subject of following work. More details of our model can be found in the manuscript [1], which will be submitted soon.

[1] D. Eilmsteiner, S. Paischer, L. Chotorlishvili, I. V. Maznichenko, S. Ostanin, I. Mertig, A. D. Rata, V. Dugaev, D.-S. Park, and A. Ernst, *Theory of magnetism in $\text{LaAl}_{1-x}\text{O}_3(001)/\text{SrTiO}_3(001)$ interfaces*, to be submitted soon.

- It may be technical, but it is unclear from the experimental point of view how the data for forward and backward field sweep was obtained. As described the samples were measured in van der Pauw configuration. Were the data averaged over different geometries to avoid misalignment voltages? And how does the potential data treatment affect the hysteretic Hall signal, which is symmetric in field and could be averaged out through averaging. Further, it is unclear how the authors technically determined the coercive field from the data.

Reply to comments: To provide additional detailed information on the AHE in the samples, we performed four-point probe Hall effect measurements in relatively low fields as schematically illustrated below. The samples were measured in PPMS using external electronics (Keithely 2400 and Keithely 2000). The magnetic fields were applied, perpendicular to the xy -plane of the samples and Hall resistance were measured in a five-loop sequence of the magnetic field (e.g. $0 \rightarrow +7 \text{ T} \rightarrow -7 \text{ T} \rightarrow +7 \text{ T} \rightarrow 0 \text{ T}$).

The main message is that the observed hysteresis in the measurements is not attributed from a geometrical contribution or data averaging. These asymmetry will not produce hysteresis, they maybe only contribute non-linear and asymmetric effects

relative to the zero field. So, it is a pure AHE intrinsic contribution of our sample. Data symmetrization could also not affect the hysteretic AHE part. Note that the data recorded in the high-field ($B \geq 13$ T) and the low-field ($B \leq 7$ T) were used for the two-band analysis and AHE, respectively. We only subtracted the OHE part from the raw data to extract the AHE part - for the case of hysteretic AHE, we did not employ the non-hysteretic AHE equation (in SI9) to describe the deconvoluted hysteretic AHE part.

Importantly, the R_{xy} difference in the forward and reverse field sweeps provides a solid evidence for the existence of the AHE hysteresis as presented in Figs. 5g,h of the revised manuscript. This technical information for new measurements has been included in the method section of the revised manuscript.

As for the second question, the coercive field (B_C) of the hysteresis of the deconvoluted R_{xy} (AHE) part were determined by using a conventional way, $B_C = (|+B_C| + |-B_C|)/2$.

- Finally, the model applied to describe the AHE in the RH data (elaborated in the SI) is symmetric in B , while RH now has asymmetric contributions due to the observed hysteresis. Can the authors comment why the model is still valid to be used?

Reply to comments: As we already indicated in our previously revised supplementary information 9: "Note that this functional expression is only valid to describe a non-hysteretic AHE, used for fittings of the symmetized non-hysteretic Hall resistance data (Fig. S9)". Therefore, in the previous manuscript, we presented only the hysteretic R_{xy} and R_H (AHE) parts after subtracting OHE contribution to the experimental R_{xy} data (recorded from the fine-field interval measurements) without modelling. For more clarity, we have further revised the supplementary information S9 and Fig. S10.

- O-K-edge XMCD: While I appreciate the clear field swap shown in the new figure 4, I would like to ask why the feature 535eV cannot be related to oxygen-Sr hybrids? In my understanding, the total electron yield in XAS is not similarly surface sensitive as e.g. XPS, so they spectra may contain significant contributions from the substrate in this region. Can the authors clarify?

Reply to comments: Since TEY-detected XAS collects all kind of secondary electrons, XAS and XPS have a nearly identical probing depth when the Auger emission is very

small, which typically occurs for medium to heavy atoms [Frazer *et al.*, *Surface Science* 537, 161 (2003)]. As for the first question, the reviewer is right in the sense that there is certainly a contribution from the O 2p-Sr 4d hybrids to the XAS at around 534-537 eV, but the contribution is minor when compared to that of the O bands hybridized with La 5d, Al 3s and Al 3p. For example, in our O K XAS spectrum of 9uc-LAO/STO, the characteristic peak of Sr 4d at 536.5 eV cannot be discerned, instead, there is a broad minimum in the intensity. This is because the probing depth of TEY at these energies is only ~4 - 5 nm and the signal for the Sr 4d is hidden by the dominant signals originating from the LAO top layer.

- I also think the statement on the Ti XMCD originating from Ti Al defect sites should be rephrased. Obviously, XMCD is element sensitive. Hence, the authors observe XMCD originating from Ti with no doubt. Their conclusion the respective Ti ions are located in the LAO layer forming anti-sites however cannot be made based on the XMCD measurement alone. The authors should clarify this to avoid misunderstanding.

Reply to comments: We thank the reviewer and agree with his/her suggestion. Regarding the point on the Ti-Al antisite defects, we refer to the following text in the supplementary S7:

*“Intriguingly, the Ti $L_{2,3}$ -edge XMCD measurements exhibit prominent dichroism ($\Delta\rho$ of the total absorption signal) at a photon energy of 457.5 eV, as illustrated in Fig. S7c, while a broad and asymmetric dichroism feature in the O K-edge was observed at RT (Fig. S7d). The 457.5 eV sharp dichroism reflects the in-plane magnetization of the t_{2g} orbitals of the Ti^{4+} ions, with respect to the anisotropic linear dichroism (Fig. S7a). **This is consistent with our DFT results for the magnetization of Ti_{Al} antisite defect in LAO, i.e. the d_{xy} orbital magnetization of Ti_{Al} in LAO (Figs. 3c,d in the main text).** No spectral features related to Ti^{3+} ions can be observed and the spectral shape of our XMCD spectrum is very similar to the pure Ti^{4+} spectrum reported by Saluzzo *et al.* [S7]. This observation also confirms that there is minor or no impact by Ti^{3+} on the magnetism of the oxidized B-site cation deficient interface.”*

The referee is completely right. From the Ti $L_{2,3}$ XMCD, we can only say that the measured XMCD is a characteristic of Ti^{4+} and there is no magnetic signature associated with Ti^{3+} . Ti $L_{2,3}$ XMCD cannot distinguish between Ti^{4+} in STO and Ti_{Al}^{4+} antisite defect in LAO although the Ti signal is barely seen from STO.

To avoid any misunderstanding, we have removed the statement in the revised supplementary information S7.

- As for the defect dynamics, I see the argument made by the authors, but I would suggest to comment on this in the paper. It is not obvious that anti-site defect can be formed under the non-equilibrium conditions of PLD, and should hence be mentioned in the manuscript.

Reply to comments: We would agree with the reviewer comments that a defect compensation process across the interface could be rather limited under non-equilibrium/moderate growth condition in PLD. However, according to our experimental observation (XPS and STEM-EELS), it is clear for Ti out-diffusion behaviour to the Al-deficient LAO overlayers keeping an ABO_3 perovskite structure as other research groups have also observed similar Ti diffusion and cation exchange behaviours at similar interfaces [Chen, *et al. Nat. Commun.* **4**, 1371 (2013)]. No segregation of Ti/TiO₂ was observed in the Al-deficient LAO overlayer. Although all the films were grown in thermodynamically non-equilibrium states/partial relaxation, such a cation exchange near the interfaces are commonly observed. This is strongly dependent on the cation stoichiometry of oxide overlayers and the interface roughness [NaKagawa, *et al. Nat. Mater.* **5**, 204 (2006), Warusawithana, *et al. Nat. Commun.* **4**, 2351 (2013), Xu, *et al. Adv. Mater.* **29**, 1604447 (2017)]. For example, for the case of B-site cation deficient LAO/STO, Ti out-diffusion behaviour from the STO into the LAO overlayer is indeed opposite to cation diffusion events in conventional conductive (Al-rich) LAO/STO interfaces, where A-site cation (Sr) out-diffusion typically occurs [Kalabukhov *et al. PRL*, **103**, 146101 (2009), Treske *et al. APL Mater.* **2**, 012108 (2014), Xu *et al. Sci. Rep.* **6**:22410 (2016)]. Hence, this B-site cation compensation process can be facilitated by the B-site cation deficiency of the LAO overlayer across the interface.

This anti-site defect formation in the B-site cation deficient system could be further promoted by longer relaxation (e.g. longer cooling-process and post-annealing) after film growth. Thus, it would be interesting to undertake a further detailed investigation on a microscopic view of Ti diffusion behaviour and defect clustering with different relaxation process after film growth, but this is beyond the scope for the main message of the present work.

Following the reviewer's comments, we have modified the manuscript accordingly:

*“Our findings indicate that the formation of interfacial magnetism stems from the atomic charge compensation across the interface region between the B-site cation deficient LAO and STO. **Such a cation exchange/intermixing is commonly observed at a complex oxide interface, which can be strongly dependent on the stoichiometry of oxide overlayers [Ref. 23, Ref. 24, Ref. 44].** Our results indicate that the Al vacancy sites in the LAO overlayer play a crucial role in initiating the creation of secondary magnetic components and consecutive defect alignment.”*

- Finally, a quantitative estimation of a magnetic moment per defect – be it anti-site, cation vacancy, or something else should be made. I understand that exact values for each potentially involved magnetic center cannot be determined, but it should be made plausible that the measured total

moment is consistent with the discussed (superimposed) mechanisms. A similar plausibility argument was also requested by reviewer #1.

Reply to comments: Experimentally it is still challenging to quantify a magnetic moment per single defect in the B-site cation deficient LAO/STO system. Therefore, we have proposed a possible mechanism for the generation of the observed large total magnetic moment of the B-site cation defect clusters in the B-site cation deficient LAO/STO system. The proposed model for a large collective orbital moment could reflect the experimentally observed large magnetic moment of the samples, strongly dependent on the B-site cation deficiency and defect assembly in the LAO/STO system as already proposed in the manuscript:

“In other words, the total magnetic response of the B-site cation-deficient heterostructure results from an extended defect configuration at the interface, instead of isolated defects or individual layer contributions. Furthermore, the preferential orientation of the magnetic moment of the B-site cation-deficient LAO/STO system is revealed to be the out-of-plane in [001] direction (Fig. 4e) and temperature-independent between 5 K – 300 K (Fig. 4f). The observed large ferromagnetism could be related to exchange interaction of the ordered defects with a large spin wave stiffness^{35,36} and/or a collective planar orbital magnetism³⁷.”

REVIEWERS' COMMENTS:

Reviewer #1 (Remarks to the Author):

I have examined the revised manuscript. My original concerns about the balance of the manuscript concerning the generation of out-of-plane magnetism in LAO/STO and the magnetotransport effects have been dealt with. The central focus of the paper is I believe appropriately focused on the very interesting generation of magnetism through defect control during LAO growth. I recommend publication of the manuscript in its current form.

Reviewer #2 (Remarks to the Author):

The authors provide compelling evidence that "materials science" matters in understanding the origin of interfacial magnetism in LAO/STO, LAO/LAO and STO/STO bilayers. Namely, the B-site occupancy is a controlling factor. To me this clarifies why some groups see magnetism when others do not.

The authors were mostly responsive to the reviewers criticisms. I understand one reviewer is not compelled by the transport story. I note the abstract and the concluding paragraph of the revised text focus on the magnetic story. In my view having some transport data presented provides context for the conclusions of the magnetic study, and a means to relate the work to previous studies (many of which present only transport data and made grandiose claims about interface magnetism).

I would also like to clarify that in my first review I did not say whether or not the two topics should be in one paper. The thought to separate the work into two pieces did not occur to me.

My recommendation is to publish the paper.

Reviewer #3 (Remarks to the Author):

The revised manuscript 'The emergence of magnetic ordering at complex oxide interfaces tuned by defects' by Park et al. shows further improvements as compared to the previous version. Also, I can mostly accept the arguments given by the authors in their response letter. Even though I think the reviewing process has revealed that the manuscript is going to cause some controversy in the research community, I think that the manuscript should be published as it presents a critical mass of very interesting data and results.

As I mentioned earlier, I personally still see this manuscript as a study that revealed highly novel insights into the magnetic properties of STO and LAO thin films, while the implications for the LAO/STO interface 'will still be discussed across the research community. At this point, however, this scientific debate can be made within the community – the manuscript describes in detail under which assumptions the conclusions were made. As such, it adds a valuable new perspective to the field and should be published.

Final minor remarks:

- I guess it would be good to share what is the 'conventional way' of determining BC – I guess the authors use the zero-crossing and average. It might not hurt to add this information to the experimental part.
- I would also mention that the hysteretic Hall effect was observed in the raw data of the Hall effect and no anti-symmetrization was applied. If it was not van der Pauw configuration, this should be

mentioned in the methods part, too (what is sketched in the response letter is not vdP, at least for Vxx).

We would like to thank the reviewers for recommending the paper for publication, and for their careful assessment and constructive inputs of the manuscript, which without any doubt lead to the improvement of the manuscript.

Reviewers #1 & #2 (Remarks to the Author):

We would like to thank the reviewers for their constructive feedback and for the recommendation to publish this work.

Reviewer #3 (Remarks to the Author):

We thank the reviewer 3 for his/her minor comments and the recommendation to publish this work. The manuscript have now been revised in respond to the following minor comments/suggestions.

- I guess it would be good to share what is the 'conventional way' of determining BC – I guess the authors use the zero-crossing and average. It might not hurt to add this information to the experimental part.

Response to the reviewer: We thank the reviewer for the comment and the manuscript has been revised accordingly. We have added the following information on the determination of the coercive field (B_C) of the hysteretic R_{xy} (AHE) parts into the Methods section of the revised manuscript:

“Coercive fields, H_C , of the extracted hysteretic 5 K R_{xy} (AHE) parts were determined by $B_C = (|+B_C| + |-B_C|)/2$, where the values of $+B_C$ and $-B_C$ are $+45 \pm 5$ mT and -49 ± 4 mT, respectively. Each B_C was obtained from a zero-crossing field, where the R_{xy} (AHE)- B hysteresis loop crosses zero R_{xy} during the field apply.”

- I would also mention that the hysteretic Hall effect was observed in the raw data of the Hall effect and no anti-symmetrization was applied. If it was not van der Pauw configuration, this should be mentioned in the methods part, too (what is sketched in the response letter is not vdP, at least for V_{xx}).

Response to the reviewer: Following the reviewer's suggestion, we have included the measurement method in the Methods section of the revised manuscript:

“To provide additional detailed information on the determination of the AHE in samples, we performed Hall measurements in a Hall bar-type configuration using the rectangular-shaped samples, i.e. charge carriers flow parallel to the single longitudinal channel and the transverse Hall voltages of the opposite sides of the channel are measured. The magnetic fields were applied, perpendicular to the xy -plane of the samples and the Hall resistance was measured in a five-loop sequence of the magnetic field (e.g. $0 \rightarrow +7 \text{ T} \rightarrow -7 \text{ T} \rightarrow +7 \text{ T} \rightarrow 0 \text{ T}$) with refine field intervals ($1 - 2 \text{ mT/sec}$). Both the high-field ($B \geq 13 \text{ T}$) and low-field ($B \leq 7 \text{ T}$) data were used for the two-band and AHE analyses, respectively.”